# Probability Distributions Computed by Autoregressive Transformers

**Andy Yang**[1]  **Anej Svete**[2]  **Jiaoda Li**[2]  **Anthony Widjaja Lin**[3,4]  **Jonathan Rawski**[5]
**Ryan Cotterell**[2]  **David Chiang**[1]

[1]University of Notre Dame, USA    [2]ETH Zürich, Switzerland
[3]Max-Planck Institute for Software Systems, Germany
[4]University of Kaiserslautern-Landau, Germany    [5]San José State University, USA

## Abstract

Most expressivity results for transformers treat them as language recognizers—devices that accept or reject strings—rather than as they are used in practice: as language models that generate strings autoregressively and probabilistically. We characterize the probability distributions that transformer language models can express. We show that making transformer language recognizers autoregressive can sometimes increase their expressivity, and that making them probabilistic can break equivalences that hold in the non-probabilistic case. Our overall contribution is to tease apart what functions transformers are capable of expressing in their most common use case as language models.

## 1 Introduction

Most work studying transformer expressivity, that is, what classes of computations transformers can perform, treats them as *language recognizers*, where the input is a string and the output is a binary classification: true if the string is accepted and false otherwise (Strobl et al., 2024). However, the most common practical use of transformers is as *language models*, which differ in two ways: first, the input is a prefix of a string, and the output is a prediction of the next symbol; second, the prediction is a probability distribution rather than a binary decision. Such probability distributions, when estimated from large text corpora, have enabled a wide range of applications in natural language processing and beyond. This paper focuses on a fundamental question: which previous findings on transformer expressivity carry over from the language recognition setting to the language modeling setting? On one hand, positive answers validate the utility of previous results on language recognition when studying language models. On the other, negative answers further underscore the necessity of studying language models *qua* language models.

In order to develop a formal theory of transformers as language models, we introduce two distinctions: *unweighted* (or equivalently, *Boolean-weighted*) versus *real-weighted* computation and *classifiers*, which map a complete string to a value, versus *autoregressors*, which map each prefix to a distribution over the next token. This four-way distinction is visualized in Fig. 1. Using this terminology, most theoretical work on transformer expressivity (e.g. Yang et al., 2024; Jerad et al., 2025) focuses on Boolean-weighted classifiers, while practical applications use transformers as real-weighted autoregressors. This work investigates whether established expressivity results remain valid when moving from Boolean-weighted to real-weighted transformers, and from classifier settings to autoregressive ones.

We answer these questions for several variants of transformers (see Fig. 1). Yang et al. (2024) proved that strictly-masked rightmost unique-hard attention transformers (UHATs), as Boolean classifiers, recognize the same languages as linear temporal logic (LTL) and counter-free automata. Jerad et al. (2025) proved that *leftmost* UHATs, as Boolean classifiers, recognize the same languages as a fragment of LTL, called in our notation TL[**P**]. Li and Cotterell (2025) proved that softmax attention transformers (SMATs) with fixed precision, as Boolean classifiers, recognize the same class of languages. These results carry over easily to real classifiers (Cor. 5.2), with the caveat that there are two commonly-used weighted analogues of counter-free automata, deterministic and nondeterministic.

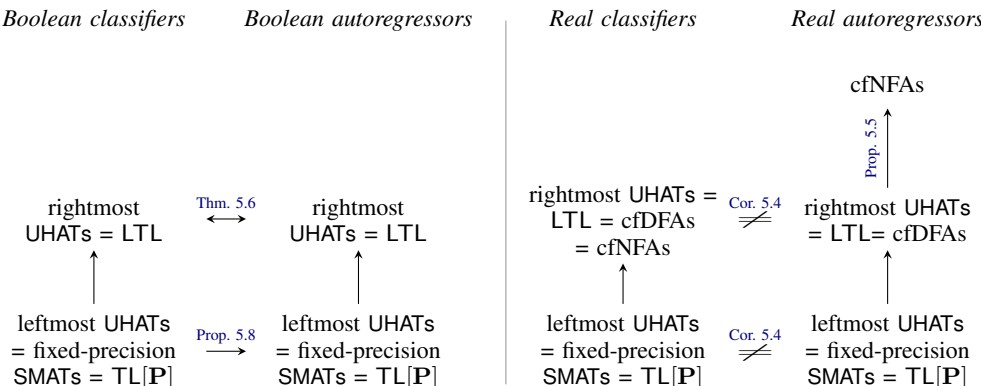

Figure 1: In the Boolean semiring, equivalences from the literature (Yang et al., 2024; Jerad et al., 2025; Yang et al., 2025) carry over from classifiers to autoregressors; however, sometimes autoregressors are more expressive than classifiers. In the real semiring, LTL and counter-free DFAs and NFAs become less expressive than counter-free NFAs, and rightmost UHATs are only as expressive as the former. Key: → strict inclusion, ↔ equivalence. ≠ incomparable.

We show that surprisingly, these two diverge in the real-weighted setting, despite being equivalent in the Boolean setting. With real weights, UHATs are only equivalent to counter-free DFAs.

This caveat notwithstanding, we may use LTL and TL[**P**] to draw conclusions about the transformer variants listed above. First, real classifiers define some weighted languages that real autoregressors do not, and vice versa (Cor. 5.4). To pinpoint more precisely how classifiers and autoregressors differ, we turn to Boolean weights. Here, UHAT classifiers and autoregressors are equivalent (Thm. 5.6). But for leftmost UHATs and fixed-precision SMATs, autoregressors are strictly more expressive than classifiers (Prop. 5.8).

Similarly, Yang et al. (2025) considered SMATs with fixed precision but arbitrary precision inside attention. As Boolean classifiers, such transformers are exactly equivalent to a temporal logic extended with counting operators. But here we show that, as autoregressors, they become slightly more powerful (for a given fixed depth).

Our results largely validate existing results on expressivity of transformers as language recognizers. In many cases, they allow us to transfer results on language recognizers to language models. For example, since UHATs as language recognizers cannot recognize PARITY (Hahn, 2020), neither can UHATs as language models. But our results also give good reasons to be cautious about presuming that results on language recognizers directly apply to language models as well.

In §2, we define notational preliminaries. We then (§3) define the classes of transformers we consider and how they can be used as classifiers and as autoregressors. We do this by introducing the notion of a **state encoder**, and in §4 show how two other formalisms, deterministic finite automata (DFAs) and linear temporal logic (LTL), can also be seen as state encoders and therefore can be used as classifiers and autoregressors. Then (§5), using LTL, we investigate the expressive power of transformers as both classifiers and autoregressors, yielding the results shown in Fig. 1.

## 2  PRELIMINARIES

Throughout this paper, we work with weighted languages. We define some key concepts here, but for a more detailed introduction, see the handbook chapter by Droste and Kuich (2009).

Let $\Sigma$ be an **alphabet**, that is, a finite, non-empty set of **symbols**, and let $\Sigma^*$ be the set of strings over $\Sigma$. We often augment $\Sigma$ with start and end symbols BOS and EOS, but never consider BOS or EOS to belong to $\Sigma$. For any string $\boldsymbol{w} = w_1 \cdots w_n$, we write the length of $\boldsymbol{w}$ as $|\boldsymbol{w}| = n$. We write $\boldsymbol{w}_{<i} = w_1 \cdots w_{i-1}$ and $\boldsymbol{w}_{\leq i} = w_1 \cdots w_i$. We write $\epsilon$ for the empty string.

We think of weights and probabilities as elements of **semirings**, an abstraction of the usual addition and multiplication operations that allows results and algorithms to apply generically to multiple settings. A semiring $\mathbb{K}$ has an addition operation $\oplus$, additive identity $\mathbf{0}$, multiplication operation $\otimes$, and multiplicative identity $\mathbf{1}$. The two semirings we focus on in this paper are the **(extended nonnegative) real semiring** $\overline{\mathbb{R}}_{\geq 0}$, which contains all nonnegative real numbers and $+\infty$, and in which $\oplus$ and $\otimes$ are real addition and multiplication; and the **Boolean semiring** $\mathbb{B}$, in which $\oplus$ is disjunction ($\vee$), $\mathbf{0}$ is false ($\bot$), $\otimes$ is conjunction ($\wedge$), and $\mathbf{1}$ is true ($\top$).

A **weighted language** (also called a **formal power series**) is a function $S \colon \Sigma^* \to \mathbb{K}$. When $\mathbb{K}$ is complete, that is, when $\mathbb{K}$ is closed under infinite summations, as $\overline{\mathbb{R}}_{\geq 0}$ and $\mathbb{B}$ are, we call a weighted language **normalized** if $\sum_{\boldsymbol{w} \in \Sigma^*} S(\boldsymbol{w}) = \mathbf{1}$.

For sets $X$ and $Y$, we write $Y^X$ for the set of functions from $X$ to $Y$, and $2^X$ for the power set of $X$. For any proposition $\phi$, we write $\mathbb{I}\{\phi\}$ to be 1 if $\phi$ is true and 0 if $\phi$ is false.

## 3 TRANSFORMER LANGUAGE MODELS

In this section, we recall the definition of transformers that we will use throughout most of this paper. We also distinguish between two ways that transformers (and other formalisms) can be used to define weighted languages.

### 3.1 UNIQUE HARD ATTENTION TRANSFORMERS

Following Yang et al. (2024), we use **unique-hard attention transformers** (UHATs), specifically, with rightmost-hard attention, strict future masking, and no position embeddings. We give a definition of strictly masked rightmost-hard attention here; for a definition of the rest of the network, see, for example, the survey by Strobl et al. (2024).

The attention function receives a sequence of query vectors $\mathbf{q}^{(i)} \in \mathbb{R}^{d_k}$, key vectors $\mathbf{k}^{(j)} \in \mathbb{R}^{d_k}$, and value vectors $\mathbf{v}^{(j)} \in \mathbb{R}^d$, all of which are column vectors, for $i, j \in [n]$. At each position $i$, it computes a sequence of vectors

$$\text{Att}\left((\mathbf{q}^{(i)})_{i \in [n]}, (\mathbf{k}^{(j)})_{j \in [n]}, (\mathbf{v}^{(j)})_{j \in [n]}\right) = (\mathbf{c}^{(i)})_{i \in [n]} \tag{1}$$

where

$$
\begin{aligned}
a_i(j) &= \mathbf{q}^{(i)} \cdot \mathbf{k}^{(j)} && \text{is an attention score for each position } j, \\
a_i^* &= \max_{j < i} a_i(j) && \text{is the maximum attention score,} \\
j_i &= \max\{j < i \mid a_i(j) = a_i^*\} && \text{is the rightmost maximum-scoring position, and} \\
\mathbf{c}^{(i)} &= \begin{cases} \mathbf{v}^{(j_i)} & \text{if } i > 0 \\ \mathbf{0} & \text{if } i = 0 \end{cases} && \text{is the attention output.}
\end{aligned}
$$

A transformer $\mathcal{T}$ consists of a word embedding $\text{Emb} \colon \Sigma \to \mathbb{R}^d$, followed by a composition of attention functions and feed-forward networks, allowing Layernorm but no using positional encodings. We defer the definitions because they are standard, and we primarily will refer to the expressive equivalence of these transformers and different formal logics as shown by Yang et al. (2024); Jerad et al. (2025); Li and Cotterell (2025). Given an input string $\boldsymbol{w} = w_1 \cdots w_n$, a transformer $\mathcal{T}$ prepends a symbol $w_0 = \text{BOS}$ and computes a sequence of states $\mathcal{T}(\boldsymbol{w}) = (\mathbf{h}^{(0)}, \ldots, \mathbf{h}^{(n)})$, where $\mathbf{h}^{(i)} \in \mathbb{R}^d$ is the state after reading $w_i$. There are at least two ways to use $\mathcal{T}$ to define a weighted language, which we describe below.[1]

### 3.2 CLASSIFIERS

The first way that a transformer can define a weighted language is as a **classifier**.

---

[1] A third intermediate way would be to multiply the weights at each position like an autoregressive model, but not to pass the output symbol at each position autoregressively to the input at the next position. Although interesting in its own right, it has not, to our knowledge, been used with any neural sequence models, and we do not explore this style of model here.

**Definition 3.1.** *A UHAT **classifier** is a pair $C = (\mathcal{T}, c)$, where $\mathcal{T}: \Sigma^* \to (\mathbb{R}^d)^*$ is a UHAT and $c: \mathbb{R}^d \to \mathbb{K}$ outputs a scalar weight at the last position only:*

$$C(\boldsymbol{w}) = c(\mathcal{T}(\boldsymbol{w})_n). \tag{2}$$

For the Boolean semiring ($\mathbb{K} = \mathbb{B}$), we accept a string iff the transformer outputs $\top$ at the last position. For example, the output function could be $c(\mathbf{y}) = \mathbb{I}\{\mathbf{w} \cdot \mathbf{y} + b \geq 0\}$, where $\mathbf{w} \in \mathbb{R}^d$ and $b \in \mathbb{R}$ are parameters. This is the setup used for binary classification with a transformer encoder (Devlin et al., 2019) and in most theoretical papers on transformer expressivity.

### 3.3 AUTOREGRESSIVE MODELS

The second way for a transformer to define a weighted language is as an **autoregressive model**, or an **autoregressor** for short (by analogy with *classifier*). An autoregressor pairs a UHAT encoder with an output function $a: \mathbb{R}^d \to \mathbb{K}^{\Sigma \cup \{\text{EOS}\}}$, which outputs at each position a weight distribution for the next symbol, including EOS. In the real semiring ($\mathbb{K} = \overline{\mathbb{R}}_{\geq 0}$), a typical example of such an output function is $r(\mathbf{h}) = \text{softmax}(\mathbf{W}\mathbf{h} + \mathbf{b})$.

To line up with the more familiar notation of conditional probability distributions, we write, for all $\sigma \in \Sigma \cup \{\text{EOS}\}$,

$$\text{Pr}_A(\sigma \mid \boldsymbol{w}_{\leq i}) = r(\mathcal{T}(\boldsymbol{w})_i)(\sigma). \tag{3}$$

This is well-defined because $\mathcal{T}(\boldsymbol{w})_i$ depends only on $\boldsymbol{w}_{\leq i}$, that is, $\boldsymbol{w}_{\leq i} = \boldsymbol{w}'_{\leq i} \iff \mathcal{T}(\boldsymbol{w})_i = \mathcal{T}(\boldsymbol{w}')_i$. As suggested by this notation, we want $\text{Pr}_A(\cdot \mid \boldsymbol{u})$ to be a probability distribution over $\Sigma \cup \{\text{EOS}\}$. But we impose a stronger condition. First, we extend $\text{Pr}_A(\sigma \mid \boldsymbol{u})$ to the probability distribution of suffixes given (possibly empty) prefixes:

$$\text{Pr}_A(\boldsymbol{v} \mid \boldsymbol{u}) = \left( \bigotimes_{i=1}^{|\boldsymbol{v}|} \text{Pr}_A(v_i \mid \boldsymbol{u}\boldsymbol{v}_{<i}) \right) \otimes \text{Pr}_A(\text{EOS} \mid \boldsymbol{u}\boldsymbol{v}) \tag{4}$$

$$\text{Pr}_A(\boldsymbol{w}) = \text{Pr}_A(\boldsymbol{w} \mid \epsilon). \tag{5}$$

Then we require that every such distribution sums to one:

**Definition 3.2.** *A UHAT **autoregressor** over a complete semiring $\mathbb{K}$ is a pair $A = (\mathcal{T}, r)$, where $\mathcal{T}: \Sigma^* \to (\mathbb{R}^d)^*$ is a UHAT, and $r: \mathbb{R}^d \to \mathbb{K}^{\Sigma \cup \{\text{EOS}\}}$ is a function such that for all $\boldsymbol{u} \in \Sigma^*$, using the notation of Eqs. (3) and (4),*

$$\bigoplus_{\boldsymbol{v} \in \Sigma^*} \text{Pr}_A(\boldsymbol{v} \mid \boldsymbol{u}) = \mathbf{1}. \tag{6}$$

This implies that:

- An autoregressor generates strings symbol by symbol. That is, for all prefixes $\boldsymbol{u}$,

$$\bigoplus_{\sigma \in \Sigma \cup \{\text{EOS}\}} \text{Pr}_A(\sigma \mid \boldsymbol{u}) = \mathbf{1}. \tag{7}$$

- An autoregressor does not have any dead ends or endless loops. That is, for all prefixes $\boldsymbol{u}$,

$$\bigotimes_{i=1}^{n} \text{Pr}_A(u_i \mid \boldsymbol{u}_{<i}) \neq \mathbf{0} \implies \text{Pr}_A(\boldsymbol{u}\boldsymbol{v}) \neq \mathbf{0} \text{ for some suffix } \boldsymbol{v}. \tag{8}$$

- An autoregressor defines a normalized weighted language.

## 4 OTHER FORMALISMS

We can analogously use other formalisms to define classifier or autoregressive models. We generalize from transformers to other formalisms by means of the following notion.

**Definition 4.1.** *A **state encoder** is a function that sends a string $w_1 \cdots w_n$ to a sequence of states $q_0, \ldots, q_n \in Q$ (where $Q$ is a finite or infinite set of states) such that $q_i$ depends only on $\boldsymbol{w}_{\leq i}$.*

Like transformers, any state encoder can be equipped with an output function $c: Q \to \mathbb{K}$ to give a classifier model (as in Def. 3.1) or $r: Q \to \mathbb{K}^{\Sigma \cup \{\text{EOS}\}}$ (as in Def. 3.2) to give an autoregressor.

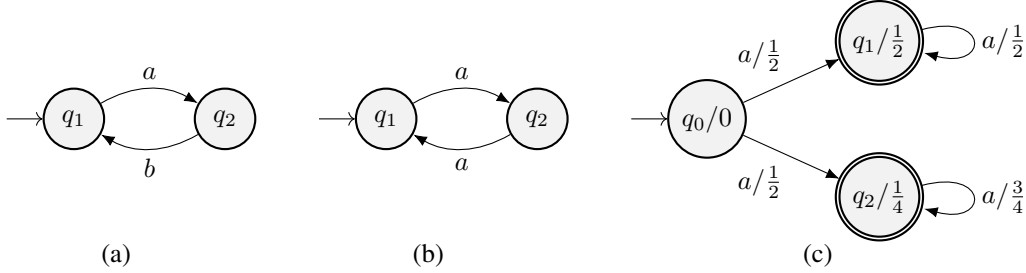

Figure 2: (a) A DFA that is counter-free (with $k = 2$). (b) A DFA that is not counter-free, because for all $k$, the strings $a^k$ and $a^{k+1}$ have opposite actions. (c) A counter-free weighted NFA that has no equivalent weighted DFA (Prop. 5.5).

### 4.1 FINITE AUTOMATA

We give brief definitions of weighted and counter-free deterministic finite automata. For a fuller treatment, please see the handbook chapter by Droste and Kuske (2021) and the monograph by McNaughton and Papert (1971).

**Definition 4.2.** *A **deterministic finite automaton** (DFA) is a tuple $M = (\Sigma, Q, \delta, \iota)$, where*

- $\Sigma$ *is an alphabet,*
- $Q$ *is a finite set of **states**,*
- $\delta \colon Q \times \Sigma \to Q$ *is a **transition function**, and*
- $\iota \in Q$ *is the **initial** state.*

*We extend $\delta$ to a mapping $\delta^* \colon Q \times \Sigma^* \to Q$ such that:*

$$\begin{aligned} \delta^*(q, \epsilon) &= q \\ \delta^*(q, \sigma \boldsymbol{w}) &= \delta^*(\delta(q, \sigma), \boldsymbol{w}). \end{aligned} \tag{9}$$

A DFA $M$ defines a state encoder

$$\begin{aligned} M &\colon \Sigma^* \to Q^* \\ M(\boldsymbol{w})_i &= \begin{cases} \iota & i = 0 \\ \delta^*(\iota, w_1 \cdots w_i) & 0 < i \le n. \end{cases} \end{aligned} \tag{10}$$

A DFA with classifier outputs in the Boolean semiring is the same as the standard definition of a DFA: the states that output $\top$ are the accept states, and the states that output $\bot$ are the reject states. A DFA with autoregressive outputs in the real semiring is the same as the standard definition of a weighted DFA: when it is in state $q$, the next input symbol $\sigma$ determines both the next state $\delta(q, \sigma)$ as well as the symbol weight $r(q)(\sigma)$. Moreover, each state has an accepting weight $r(q)(\text{EOS})$.

In this paper, we are only interested in the following subclass of finite automata called **counter-free automata**, which we abbreviate as cfDFAs.

**Definition 4.3.** *We say that a DFA with transition function $\delta$ is **counter-free** if there exists some $k$ such that for all states $q$ and all strings $\boldsymbol{w}$, we have $\delta^*(q, \boldsymbol{w}^k) = \delta^*(q, \boldsymbol{w}^{k+1})$.*

Examples of counter-free and non-counter-free DFAs are shown in Fig. 2ab.

Counter-free **nondeterministic finite automata** (NFAs), in which a state can have more than one outgoing transition with the same symbol are equivalent to counter-free DFAs (McNaughton and Papert, 1971). For a definition and proof of equivalence, please see App. C.

### 4.2 LINEAR TEMPORAL LOGIC

We give a brief definition of linear temporal logic and its fragments. For a fuller treatment, please see the article by Goranko and Rumberg (2025).

**Definition 4.4.** *The formulas of past* LTL *are defined by the grammar*

$$\phi ::= \neg\phi_1 \mid \phi_1 \wedge \phi_2$$

| $\mid \sigma$ | $\sigma \in \Sigma$ |
| $\mid$ BOS | *Beginning of string* |
| $\mid \mathbf{Y}\phi_1$ | *Yesterday* |
| $\mid \mathbf{H}\phi_1$ | *Historically* |
| $\mid \phi_1 \mathbf{S} \phi_2$ | *Since* |

*Formulas $\top$ (true), $\bot$ (false), $\phi_1 \vee \phi_2$, $\phi_1 \leftrightarrow \phi_2$, and so on, can be defined as syntactic sugar in terms of the above. The temporal operator $\mathbf{P}\phi$ (which holds iff $\phi$ was Previously true at some time) can be defined as $\mathbf{H}\phi = \neg(\mathbf{P}(\neg\phi))$.*

*The semantics of formulas is given by the relation $\boldsymbol{w}, i \models \phi$ ("$\boldsymbol{w}$ satisfies $\phi$ at position $i$"), defined as follows:*

$$\boldsymbol{w}, i \models \neg\phi_1 \iff \boldsymbol{w}, i \not\models \phi_1 \tag{11a}$$

$$\boldsymbol{w}, i \models \phi_1 \wedge \phi_2 \iff \boldsymbol{w}, i \models \phi_1 \text{ and } \boldsymbol{w}, i \models \phi_2 \tag{11b}$$

$$\boldsymbol{w}, i \models \text{BOS} \iff i = 0 \tag{11c}$$

$$\boldsymbol{w}, i \models \sigma \iff w_i = \sigma \tag{11d}$$

$$\boldsymbol{w}, i \models \mathbf{Y}\phi_1 \iff i > 0 \text{ and } \boldsymbol{w}, i-1 \models \phi_1 \tag{11e}$$

$$\boldsymbol{w}, i \models \mathbf{H}\phi_1 \iff \boldsymbol{w}, j \models \phi_1 \text{ for all } j \leq i \tag{11f}$$

$$\boldsymbol{w}, i \models \phi_1 \mathbf{S} \phi_2 \iff (\boldsymbol{w}, j \models \phi_2 \text{ for some } j \leq i) \text{ and } (\boldsymbol{w}, j' \models \phi_1 \text{ for all } j < j' \leq i). \tag{11g}$$

*We write $\boldsymbol{w} \models \phi$ as shorthand for $\boldsymbol{w}, |\boldsymbol{w}| \models \phi$.*

For any set of operators $\mathcal{O} \subseteq \{\mathbf{Y}, \mathbf{H}, \mathbf{S}\}$, we write $\mathsf{TL}[\mathcal{O}]$ for the set of formulas using only operators in $\mathcal{O}$. Thus past $\mathsf{LTL} = \mathsf{TL}[\mathbf{Y}, \mathbf{S}]$. Given a tuple of formulas $\Phi = (\phi_1, \ldots, \phi_m)$, we can define a state encoder (typically we will use $\Phi$ to refer to both the state encoder and tuple of formulas to make their connection more obvious)

$$\Phi \colon \Sigma^* \to (\mathbb{B}^m)^*$$
$$\Phi(\boldsymbol{w})_i = (\mathbb{I}\{\boldsymbol{w}, i \models \phi_1\}, \ldots \mathbb{I}\{\boldsymbol{w}, i \models \phi_m\}). \tag{12}$$

We typically write $(\Phi, r)$ for the autoregressor using the state encoder induced by $\Phi$.

Droste and Gastin (2019) define a weighted first-order logic, with several variations corresponding to several subclasses of weighted counter-free automata. Mandrali and Rahonis (2013; 2015) do the same for LTL. Both of these logics have, roughly speaking, four layers: (1) a core Boolean logic, (2) weights conditioned on formulas, (3) products over positions, and (4) addition and sums over positions. This is similar to our framework, which has (1) a core Boolean logic, (2) classifier output functions that can choose weights conditioned on formulas, and (3) autoregressive output functions that can also compute products over positions.

## 5 EXPRESSIVITY RESULTS

Previous results have shown that UHATs, LTL, and cfDFAs are equivalent in terms of language recognition. In §5.1, we use the results to show that these formalisms are also equivalent as weighted classifiers and as autoregressors.

Next, we compare the expressivity of classifier versus autoregressive models. Given the equivalence of the above formalisms, we will mainly discuss LTL. In the real semiring (§5.2), LTL classifiers define exactly the aperiodic step functions (defined below), which are less expressive than LTL autoregressors. And LTL autoregressors, in turn, are equivalent to counter-free DFA autoregressors and less expressive than weighted counter-free NFAs.

In the Boolean semiring, LTL classifiers and autoregressors are equivalent, which is the main result of §5.3.1. However, when we consider fragments of LTL, this equivalence breaks down, and autoregressors may become more expressive than classifiers (§5.3.2). Similarly, in the temporal logic with counting $\mathsf{TL}[\overleftarrow{\#}]$ and the programming language C-RASP (Yang and Chiang, 2024), autoregressors are more expressive than classifiers (§5.3.3).

## 5.1 State Encoders

We say that two state encoders $\tau_1 \colon \Sigma^* \to Q_1^*$ and $\tau_2 \colon \Sigma^* \to Q_2^*$ are **equivalent** if there is a bijection $f \colon Q_1 \to Q_2$ such that for all $\boldsymbol{w} \in \Sigma^*$, $f(\tau_1(\boldsymbol{w})) = \tau_2(\boldsymbol{w})$.

**Theorem 5.1.** *UHATs,* LTL, *and cfDFAs define equivalent state encoders.*

*Proof.* See App. A. The proof is an adaptation of existing results (Yang et al., 2024; Schützenberger, 1965; McNaughton and Papert, 1971; Kamp, 1968) connecting UHATs, LTL and cfDFAs as language recognizers. □

The following is an immediate consequence of Thm. 5.1 and the definitions of classifier and autoregressive models.

**Corollary 5.2.** *UHATs,* LTL, *and cfDFAs as classifier models define the same weighted languages. Similarly when they are used as autoregressive models.*

*Proof.* By the previous theorem, all these formalisms define equivalent state encoders. Therefore there exist output functions with which they define the same weighted languages. □

## 5.2 Real classifiers and autoregressors

In this section, we consider weights in the real semiring. We characterize what weighted languages can be expressed, first by real classifiers, then by real autoregressors.

**Definition 5.1.** *An **aperiodic step function** (Droste and Gastin, 2008) is a weighted language $S \colon \Sigma^* \to \mathbb{K}$ such that $S(\boldsymbol{w}) = \bigoplus_{i=1}^{m} k_i \otimes \mathbb{I}\{\boldsymbol{w} \in L_i\}$ where $k_1, \ldots, k_m \in \mathbb{K}$ are constants and $L_1, \ldots, L_m$ are aperiodic, that is, counter-free, regular languages.*

**Proposition 5.3.** *An* LTL *classifier defines the aperiodic step functions.*

*Proof.* Given any aperiodic step function as defined above, we can write, for each $L_i$, an LTL formula $\phi_i$. Then we can write a classifier output function $c(\mathbf{h}) = \bigoplus_{i=1}^{m} k_i \otimes h_i$.

Conversely, given an LTL classifier consisting of a tuple of formulas $(\phi_1, \ldots, \phi_m)$ and an output function $c(\mathbf{h})$, for every $\mathbf{h} \in \mathbb{B}^{[m]}$, write the formula $\phi_{\mathbf{h}} = \bigwedge_{i=1}^{m}(\phi_i \leftrightarrow h_i)$. For every $\mathbf{h}$, let $L_{\mathbf{h}}$ be the language defined by $\phi_{\mathbf{h}}$. Then the weighted language can be written as the step function $S(\boldsymbol{w}) = \bigoplus_{\mathbf{h} \in \mathbb{B}^{[m]}} c(\mathbf{h}) \otimes \mathbb{I}\{\boldsymbol{w} \in L_{\mathbf{h}}\}$. □

The following easy corollary of Prop. 5.3 shows that autoregressors and classifiers are incomparable. It makes use of weighted regular expressions (Sakarovitch, 2009), in which the expression $\sigma$ (for any $\sigma \in \Sigma$) matches symbol $\sigma$ with weight $\mathbf{1}$, while the expression $k$ (for any $k \in \mathbb{K}$) matches $\epsilon$ with weight $k$.

**Corollary 5.4.** *In the real semiring: (a) The weighted language $(\frac{1}{2}a)^*$ is expressible by an* LTL *autoregressor, but not by any* LTL *classifier. (b) The language $(1a)^*$ is expressible by an* LTL *classifier but not any* LTL *autoregressor.*

*Both (a) and (b) hold with* LTL *replaced by* TL[**H**].

*Proof.* The first language has an infinite number of string weights, but an aperiodic step function can only output a finite number of different weights. On the other hand, it is easy to write an LTL (or TL[**H**]) autoregressor to recognize this. The second language can easily be expressed by a classifier assigning weight $1$ to every string of zero or more $a$'s, but is not expressible by any autoregressor because it is not a normalized weighted language. □

As real autoregressors, LTL formulas are equivalent to counter-free DFAs by Cor. 5.2. However, there are several nonequivalent weighted analogues of counter-free automata (Droste and Gastin, 2008), and LTL and UHAT autoregressors are only equivalent to the least powerful of these. In particular, both are less expressive than weighted counter-free NFAs.

**Proposition 5.5.** *Weighted counter-free NFAs define more weighted languages than counter-free DFA autoregressors do.*

*Proof.* See App. C. Fig. 2c shows an example of a counter-free weighted NFA that is not determinizable. □

## 5.3 BOOLEAN CLASSIFIERS AND AUTOREGRESSORS

To examine more carefully how autoregressors add expressivity, we turn to the Boolean semiring. We will see that LTL classifiers and LTL autoregressors are equivalent, but with an important caveat: with certain fragments and extension of LTL that use only a subset of the temporal operators, autoregressors can be more expressive than classifiers. These variants of LTL are particularly interesting because they have been proven to be equivalent to variants of transformers.

### 5.3.1 LTL

In the Boolean semiring, LTL classifiers and autoregressors are equivalent, but the conversion from an autoregressor to a classifier uses the $\mathbf{Y}$ and $\mathbf{H}$ operators.

**Theorem 5.6.** *For any set of operators $\mathcal{O} \subseteq \{\mathbf{Y}, \mathbf{H}, \mathbf{S}\}$:*

*(a) For any nonempty language $L$ defined by a Boolean-weighted $\mathsf{TL}[\mathcal{O}]$ classifier, there exists a Boolean-weighted $\mathsf{TL}[\mathcal{O}]$ autoregressor defining the same language $L$.*

*(b) For any language $L$ defined by a Boolean-weighted $\mathsf{TL}[\mathcal{O}]$ autoregressor, there exists a Boolean-weighted $\mathsf{TL}[\mathcal{O} \cup \{\mathbf{Y}, \mathbf{H}\}]$ classifier defining the same language $L$.*

*Proof.* See App. B.3 for the full proof; a proof sketch follows.

To prove (a), we need to construct an autoregressor that tests, given any position $i$ and symbol $\sigma$, whether $\boldsymbol{w}_{\leq i}\sigma$ is a prefix of some string that is accepted by the classifier. To do this, we introduce two new operators as syntactic sugar that do not increase the expressivity of the logic:

$$\boldsymbol{w} \models \mathrm{next}_\sigma(\phi) \iff \boldsymbol{w}\sigma \models \phi$$
$$\boldsymbol{u} \models \mathrm{prefix}(\phi) \iff \text{there exists } \boldsymbol{v} \in \Sigma^* \text{ such that } \boldsymbol{u}\boldsymbol{v} \models \phi.$$

The $\mathrm{next}_\sigma$ operator is what lets us hypothesize $\sigma$ as the next symbol, and the $\mathrm{prefix}$ operator is what lets us hypothesize the rest of the string.

To prove (b), we need to construct a classifier that tests whether, for every position $i$, the autoregressor predicts that $\boldsymbol{w}_{<i}$ can be followed by $\boldsymbol{w}_i$. To test the relationship between each prefix and the next symbol, we use the $\mathbf{Y}$ operator, and to do so at every position, we use the $\mathbf{H}$ operator. □

From Thm. 5.6, we can conclude that for UHATs, which are equivalent to LTL, autoregression does not add any expressivity. On other transformer variants, please see §5.3.2.

The construction that desugars $\mathrm{prefix}(\phi)$ into a formula of $\mathsf{TL}[\mathcal{O}]$ yields a formula whose size is exponential in that of $\phi$. To shed light on whether this bound is tight, we show the following.

**Proposition 5.7.** *(a) There does not exist a transformation $\mathrm{prefix}'$ such that $\mathrm{prefix}'(\phi)$ is constructible in polynomial time (in $|\phi|$) and satisfies Eq. (19) for every formula $\phi$ in $\mathsf{TL}[\mathbf{H}, \mathbf{Y}]$, unless $\mathsf{P} = \mathsf{PSPACE}$.*

*(b) Similarly for $\mathsf{TL}[\mathbf{H}]$, unless $\mathsf{P} = \mathsf{NP}$.*

*(c) Similarly for $\mathsf{TL}[\mathbf{Y}]$, unless $\mathsf{P} = \mathsf{NP}$.*

*Proof.* See App. B.4. This is a reduction from existing results on the hardness of testing whether a formula defines an empty language (Giacomo and Vardi, 2013; Fionda and Greco, 2016). □

Note that we have only shown (conditionally) that constructing $\mathrm{prefix}'(\phi)$ requires super-polynomial time; it's possible that $\mathrm{prefix}'(\phi)$ is short but difficult to construct.

### 5.3.2 FRAGMENTS OF LTL

Thm. 5.6 shows that LTL classifiers and autoregressors are equivalent, and this remains true for some fragments of LTL. But the asymmetric conditions of the theorem suggest that when the set of operators $\mathcal{O}$ lacks either $\mathbf{H}$ or $\mathbf{Y}$, Boolean autoregressors are more expressive than classifiers. In this section, we prove that this is indeed the case.

Moreover, such fragments are relevant to the study of transformers. Li and Cotterell (2025) show that fixed-precision future-masked transformers are equivalent to $\mathsf{TL}[\mathbf{P}]$, which is in turn equivalent to $\mathsf{TL}[\mathbf{H}]$. Similarly, Jerad et al. (2025) show that future-masked leftmost-hard attention transformers are also equivalent to $\mathsf{TL}[\mathbf{P}]$.

**Proposition 5.8.** *The language* $(ab)^*$ *is defined by a Boolean* $\mathsf{TL}[\emptyset]$ *autoregressor but not defined by any* $\mathsf{TL}[\mathbf{H}]$ *or* $\mathsf{TL}[\mathbf{Y}]$ *classifier.*

*Proof.* Consider the state encoder induced by the triple of formulas $\Phi = (\textsc{bos}, a, b)$ as in Eq. (12),

$$\Phi(\boldsymbol{w})_i = (\mathbb{I}\{\boldsymbol{w}, i \models \textsc{bos}\}, \mathbb{I}\{\boldsymbol{w}, i \models a\}, \mathbb{I}\{\boldsymbol{w}, i \models b\})$$

$$r\colon \mathbb{B}^3 \to \mathbb{B}^{\{a,b,\textsc{eos}\}}$$
$$r((q_{\textsc{bos}}, q_a, q_b))(a) = \top \iff q_{\textsc{bos}} = \top \text{ or } q_b = \top$$
$$r((q_{\textsc{bos}}, q_a, q_b))(b) = \top \iff q_a = \top$$
$$r((q_{\textsc{bos}}, q_a, q_b))(\textsc{eos}) = \top \iff q_{\textsc{bos}} = \top \text{ or } q_b = \top.$$

This defines $(ab)^*$.

But a formula in $\mathsf{TL}[\mathbf{Y}]$ cannot distinguish between strings that differ beyond their last $k$ symbols (for some constant $k$ depending on the formula), and for any $k$, we have $ab(ab)^{\lceil k/2 \rceil} \in (ab)^*$ but $ba(ab)^{\lceil k/2 \rceil} \notin (ab)^*$. A formula in $\mathsf{TL}[\mathbf{H}]$ is equivalent to one in $\mathsf{TL}[\mathbf{P}]$, which can only define a stutter-invariant language, that is, a language $L$ such that for all $\boldsymbol{u}, \sigma, \boldsymbol{v}$, we have $\boldsymbol{u}\sigma\boldsymbol{v} \in L \iff \boldsymbol{u}\sigma\sigma\boldsymbol{v} \in L$ (Peled and Wilke, 1997). And $(ab)^*$ is not stutter-invariant, because $ab \in (ab)^*$ but $aab \notin (ab)^*$. □

Consequently, $(ab)^*$ is definable by leftmost-hard UHATs and fixed-precision SMATs as autoregressors, but not as classifiers. However, the expressiveness added by autoregression remains limited, as $(aab)^*$ is not definable.

**Proposition 5.9.** *The language* $(aab)^*$ *is not definable by any* $\mathsf{TL}[\mathbf{H}]$ *classifier or autoregressor.*

*Proof.* See App. D. We show that to distinguish $aab$ from $aaab$, we need at least two nested $\mathbf{Y}$ operators. But the conversion from an autoregressor to a classifier (Thm. 5.6(b)) adds only one $\mathbf{Y}$, so $(aab)^*$ is not definable by any $\mathsf{TL}[\mathbf{H}]$ autoregressor. □

Consequently, $(aab)^*$ is not definable by any leftmost-hard UHAT or fixed-precision SMAT, either as autoregressors or classifiers.

### 5.3.3 Temporal Logic with Counting

Other formalisms besides the ones discussed above have been proposed for comparison with transformers. Yang et al. (2025) prove that SMATs, with fixed precision outside attention and arbitrary precision inside attention, are equivalent to a temporal logic with counting operators, $\mathsf{TL}[\overleftarrow{\#}]$. They considered the family of languages

$$L_1 = a^* \qquad (13) \qquad\qquad L_{k+1} = \begin{cases} L_k b^* & k \text{ even} \\ L_k a^* & k \text{ odd} \end{cases} \qquad (14)$$

and showed that, as Boolean classifiers, transformers with depth $k$ can recognize $L_k$ (and not $L_{k+1}$). But their experiments were on the symbol-prediction task (§6), closely related to Boolean autoregression. They showed both theoretically and experimentally that SMATs with depth $k$ can solve the symbol-prediction task for not only $L_k$, but $L_{k+2}$ (and not $L_{k+3}$). In the present framework, this discrepancy can be readily explained. Like $\mathsf{TL}[\mathbf{H}]$, the logic $\mathsf{TL}[\overleftarrow{\#}]$ lacks a $\mathbf{Y}$ operator or an equivalent. So it is more expressive as an autoregressor than as a classifier.

## 6 Related Work

Theoretical study of transformers as language models has not gone totally neglected. Hahn (2020) compared a SMAT language model with a probabilistic finite automaton for parity (strings that have an odd number of 1's). Yao et al. (2021), following previous work on RNNs, considered a transformer language model to $\epsilon$-generate a language if it assigns probability at least $\epsilon$ to each symbol in every string in the language (and no strings not in the language). They also discussed how to convert a construction for a bounded Dyck language (strings of matching parentheses up to a certain depth) from an $\epsilon$-generator to a language recognizer. Svete and Cotterell (2024) showed that average-hard

attention transformer language models can exactly express all $n$-gram language models. These studies were specialized to particular languages, or used specialized ways of comparing distributions that do not generalize in an obvious way.

Experimentally, Bhattamishra et al. (2020a) proved theoretical results on transformers as language recognizers but carried out experiments on transformer language models for the character prediction task: predict, at each position, the set of next possible symbols, that is, Boolean autoregression. This experimental setup was previously used in studies of RNNs, and has been adopted in other studies of transformers (Huang et al., 2025; Yang et al., 2025), which we discussed in §5.3.3. The idea that the sequence of output vectors of a transformer and the states of a finite automaton can be connected via the notion of a state encoder is not new; previous results on using transformers to simulate (weighted) finite automata made a similar connection (Liu et al., 2023; Rizvi-Martel et al., 2024).

## 7 DISCUSSION

We have observed settings where classifiers coincide with autoregressors, and settings where they do not. Where the two do coincide (e.g., Boolean-weighted LTL and UHATs), we can now transfer results on definability from the more well-understood world of classifiers to autoregressors. For example, PARITY is not expressible by UHAT classifiers (Hahn, 2020), and therefore not by UHAT autoregressors. Where classifiers and autoregressors do not coincide (e.g., with real weights), we do not yet have good techniques for showing inexpressibility by autoregressors, which is left for future work.

One direction for future work is to extend our results to more realistic classes of transformers by considering log-precision softmax attention or positional encodings. The former would require stronger characterizations of log-precision softmax attention transformers. The latter could utilize connections between positional encodings and numerical predicates, as discussed by Yang et al. (2024); Barcelo et al. (2024). Another future avenue is studying autoregressors enriched with chain of thought, that is, allowing the autoregressor to run for a number of intermediate steps before producing an answer. There are numerous results relating transformers with chain of thought and different classes of computational problems (Pérez et al., 2021; Merrill and Sabharwal, 2024; Bhattamishra et al., 2020b; Li et al., 2024; Nowak et al., 2024; Li and Wang, 2025; Hou et al., 2025), but only with Boolean weights. What can be said about real-weighted autoregressors with chain of thought, and the probability distributions they compute? For example, can they compute those of probabilistic Turing machines solving problems in the BPP or even PP complexity classes?

Lastly, by clarifying the relationship between classifiers and autoregressors, our results provide a principled way to interpet and build upon the various experiments that test the expressive capabilities of transformers (Weiss et al., 2018; Bhattamishra et al., 2020a; van der Poel et al., 2024; Delétang et al., 2023; Someya et al., 2024; Borenstein et al., 2024; Butoi et al., 2025, *inter alia*). For example, we are able to explain in §5.3.3 why transformers as classifiers and autoregressors exhibit different expressive capabilities in Yang et al. (2025)'s experiments. More generally, we have identified several theoretical separations between the expressive capabilities of classifiers and autoregressors in both the Boolean and real-weighted cases, which future work could probe experimentally.

## ACKNOWLEDGEMENTS

This paper developed from the findings of the working group on probability at Dagstuhl Seminar 25282, "Theory of Neural Language Models." We are grateful to the Leibniz Center for Informatics for their support. We also thank Gavin Dooley for his feedback and a correction. This material is based in part upon work supported by the US National Science Foundation under Grant No. 2502292 and the European Research Council under Grant No. 101089343. Andy Yang is supported by the US National Science Foundation Graduate Research Fellowship Program under Grant No. 2236418, and Anej Svete is supported by the ETH AI Center Doctoral Fellowship.

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

## A  EQUIVALENCE OF STATE ENCODERS

**Theorem 5.1.** *UHATs, LTL, and cfDFAs define equivalent state encoders.*

*Proof.* First we show the equivalence of state sequences defined by UHATs and LTL, and then equivalence of LTL and cfDFAs.

The essential observation (Yang et al., 2024, Lemma 22) is that the output at every position of every UHAT layer comes from a finite set $Q \subseteq \mathbb{R}^d$. So we can think of a UHAT as a function $\mathcal{T} \colon \Sigma^* \to Q^*$. For each $q \in Q$, we can construct an LTL formula $\phi_q$ such that $\mathcal{T}(\boldsymbol{w})_i = q \iff \boldsymbol{w}, i \models \phi_q$ (Yang et al., 2024, Theorems 2, 4). So there exists a tuple of LTL formulas $(\phi_q)_{q \in Q}$ that defines a state encoder equivalent to $\mathcal{T}$. Note that the state outputted by $\mathcal{T}$ on the prepended BOS symbol can be simulated using a BOS formula in the tuple.

In the other direction, for every tuple of LTL formulas $(\phi_1, \phi_2, \ldots, \phi_m)$ defining a state encoder $\Sigma^* \to \mathbb{B}^m$, there exists a UHAT $\mathcal{T} \colon \Sigma^* \to (\mathbb{R}^d)^*$ defining an equivalent state encoder. For each $\phi_k$, we construct a transformer $\mathcal{T}_k$ which outputs $\frac{1}{2}$ if $\boldsymbol{w}, i \models \phi_k$ and $-\frac{1}{2}$ otherwise (Yang et al., 2024, Theorems 1, 3). Then we can parallel-compose all the $\mathcal{T}_k$ into a single $\mathcal{T}$ (Yang et al., 2024, Lemma 25), and add an additional layer which projects the output dimensions of each $\mathcal{T}_k$ into a single output vector $\mathbb{R}^m$ such that $\mathcal{T}(\boldsymbol{w})_i = \mathbf{e}_k \iff \boldsymbol{w}, i \models \phi_k$.

The equivalence between LTL and cfDFAs can be described a little more succinctly. Given a DFA $M = (\Sigma, Q, \delta, \iota)$, for each state $q \in Q$ there exists a formula $\phi_q$ such that $\boldsymbol{w} \models \phi_q \iff \delta(\iota, \boldsymbol{w}) = q$, due to the expressive equivalence of LTL and cfDFAs (Schützenberger, 1965; McNaughton and Papert, 1971; Kamp, 1968). The tuple $(\phi_q)_{q \in Q}$ then defines a state encoder equivalent to $M$. In the other direction, given a tuple of LTL formulas $(\phi_1, \ldots, \phi_m)$, for each $k \in [m]$ there is an automaton $M_k$ that recognizes the same language as $\phi_k$. Then the Cartesian product of all the $M_k$ defines a state encoder equivalent to $(\phi_1, \ldots, \phi_m)$. $\qquad\square$

## B  AUTOREGRESSIVE MODEL PROOFS

### B.1  PROOF OF LEM. B.1

**Lemma B.1.** *There is a transformation* $\mathrm{next}_\sigma$ *from formulas of* $\mathsf{TL}[\mathcal{O}]$ *to formulas of* $\mathsf{TL}[\mathcal{O}]$ *such that for any formula* $\phi$ *of* $\mathsf{TL}[\mathcal{O}]$ *and for all* $\boldsymbol{w} \in \Sigma^*$,

$$\boldsymbol{w} \models \mathrm{next}_\sigma(\phi) \iff \boldsymbol{w}\sigma \models \phi. \tag{15}$$

Intuitively, $\mathrm{next}_\sigma$ removes a $\sigma$ on the right; in other words, $\mathrm{next}_\sigma(\phi)$ defines the right Brzozowski derivative (Brzozowski, 1964) of the language defined by $\phi$.

*Proof.* We define $\mathrm{next}_\sigma$ recursively:

$$\mathrm{next}_\sigma(\sigma) = \top \tag{16a}$$
$$\mathrm{next}_\sigma(\sigma') = \bot \qquad \text{if } \sigma' \neq \sigma \tag{16b}$$
$$\mathrm{next}_\sigma(\text{BOS}) = \bot \tag{16c}$$
$$\mathrm{next}_\sigma(\neg\phi) = \neg\mathrm{next}_\sigma(\phi) \tag{16d}$$
$$\mathrm{next}_\sigma(\phi_1 \wedge \phi_2) = \mathrm{next}_\sigma(\phi_1) \wedge \mathrm{next}_\sigma(\phi_2) \tag{16e}$$
$$\mathrm{next}_\sigma(\mathbf{Y}\phi) = \phi \tag{16f}$$
$$\mathrm{next}_\sigma(\mathbf{H}\phi) = \mathbf{H}\phi \wedge \mathrm{next}_\sigma(\phi) \tag{16g}$$
$$\mathrm{next}_\sigma(\phi_1 \mathbf{S} \phi_2) = (\mathrm{next}_\sigma(\phi_1) \wedge (\phi_1 \mathbf{S} \phi_2)) \vee \mathrm{next}_\sigma(\phi_2). \tag{16h}$$

Note that $\mathrm{next}_\sigma$ never translates a temporal operator into another temporal operator, so it translates formulas of $\mathsf{TL}[\mathcal{O}]$ into formulas of $\mathsf{TL}[\mathcal{O}]$ for any $\mathcal{O}$.

Next, we prove that $\mathrm{next}_\sigma(\phi)$ satisfies Eq. (15) by induction on the structure of $\phi$.

**Base Cases.** If $\phi = \sigma$:

$$\boldsymbol{w}, i \models \mathrm{next}_\sigma(\sigma) \overset{(16a)}{\iff} \boldsymbol{w} \models \top \tag{17a}$$
$$\overset{(11d)}{\iff} \boldsymbol{w}\sigma \models \sigma. \tag{17b}$$

If $\phi = \sigma'$ for $\sigma' \neq \sigma$:

$$w \models \text{next}_\sigma(\sigma') \overset{(16\text{b})}{\iff} w \models \bot \tag{17c}$$

$$\overset{(11\text{d})}{\iff} w\sigma \models \sigma'. \tag{17d}$$

Similarly, if $\phi = \text{BOS}$:

$$w \models \text{next}_\sigma(\text{BOS}) \overset{(16\text{c})}{\iff} w \models \bot \tag{17e}$$

$$\overset{(11\text{c})}{\iff} w\sigma \models \text{BOS}. \tag{17f}$$

**Inductive Cases.** If $\phi = \neg\phi_1$:

$$w \models \text{next}_\sigma(\neg\phi_1) \overset{(16\text{d})}{\iff} w \models \neg\text{next}_\sigma(\phi_1) \tag{18a}$$

$$\overset{(11\text{a})}{\iff} w \not\models \text{next}_\sigma(\phi_1) \tag{18b}$$

$$\overset{\text{ind. hyp.}}{\iff} w\sigma \not\models \phi_1 \tag{18c}$$

$$\overset{(11\text{a})}{\iff} w\sigma \models \neg\phi_1. \tag{18d}$$

If $\phi = \phi_1 \wedge \phi_2$:

$$w \models \text{next}_\sigma(\phi_1 \wedge \phi_2) \overset{(16\text{e})}{\iff} w \models \text{next}_\sigma(\phi_1) \wedge \text{next}_\sigma(\phi_2) \tag{18e}$$

$$\overset{(11\text{b})}{\iff} (w \models \text{next}_\sigma(\phi_1)) \wedge (w \models \text{next}_\sigma(\phi_2)) \tag{18f}$$

$$\overset{\text{ind. hyp.}}{\iff} (w\sigma \models \phi_1) \wedge (w\sigma \models \phi_2) \tag{18g}$$

$$\overset{(11\text{b})}{\iff} w\sigma \models \phi_1 \wedge \phi_2. \tag{18h}$$

If $\phi = \mathbf{Y}\phi_1$:

$$w \models \text{next}_\sigma(\mathbf{Y}\phi_1) \overset{(16\text{f})}{\iff} w \models \phi_1 \tag{18i}$$

$$\overset{(11\text{e})}{\iff} w\sigma \models \mathbf{Y}\phi_1. \tag{18j}$$

If $\phi = \mathbf{H}\phi_1$:

$$w \models \text{next}_\sigma(\mathbf{H}\phi_1) \overset{(16\text{g})}{\iff} w \models \mathbf{H}\phi_1 \wedge \text{next}_\sigma(\phi) \tag{18k}$$

$$\overset{(11\text{b})}{\iff} (w \models \mathbf{H}\phi_1) \wedge (w \models \text{next}_\sigma(\phi)) \tag{18l}$$

$$\overset{\text{ind. hyp.}}{\iff} (w \models \mathbf{H}\phi_1) \wedge (w\sigma \models \phi_1) \tag{18m}$$

$$\overset{(11\text{f})}{\iff} w\sigma \models \mathbf{H}\phi_1. \tag{18n}$$

If $\phi = \phi_1 \mathbf{S} \phi_2$:

$$w \models \text{next}_\sigma(\phi_1 \mathbf{S} \phi_2)$$

$$\overset{(16\text{h})}{\iff} w \models (\text{next}_\sigma(\phi_1) \wedge (\phi_1 \mathbf{S} \phi_2)) \vee \text{next}_\sigma(\phi_2) \tag{18o}$$

$$\overset{(11\text{a}) \text{ and } (11\text{b})}{\iff} (w \models \text{next}_\sigma(\phi_1) \wedge (w \models \phi_1 \mathbf{S} \phi_2)) \vee (w \models \text{next}_\sigma(\phi_2)) \tag{18p}$$

$$\overset{\text{ind. hyp.}}{\iff} ((w\sigma \models \phi_1) \wedge (w \models \phi_1 \mathbf{S} \phi_2)) \vee (w\sigma \models \phi_2) \tag{18q}$$

$$\overset{(11\text{g})}{\iff} w\sigma \models \phi_1 \mathbf{S} \phi_2. \tag{18r}$$

$$\square$$

## B.2 PROOF OF LEM. B.2

**Lemma B.2.** *There is a transformation* prefix *from formulas of* $\mathsf{TL}[\mathcal{O}]$ *to formulas of* $\mathsf{TL}[\mathcal{O}]$ *such that for any formula $\phi$ of $\mathsf{TL}[\mathcal{O}]$ and for all $\boldsymbol{u} \in \Sigma^*$,*

$$\boldsymbol{u} \models \text{prefix}(\phi) \iff \text{there exists } \boldsymbol{v} \in \Sigma^* \text{ such that } \boldsymbol{u}\boldsymbol{v} \models \phi. \tag{19}$$

*Proof.* Given a formula $\phi$ of $\mathsf{TL}[\mathcal{O}]$, let $\text{cl}(\phi)$ be the set of all subformulas of $\phi$ (including $\phi$ itself). Construct a DFA $M_\phi = (2^{\text{cl}(\phi)}, \Sigma, \delta, \iota, F)$, where

$$\iota = \{\chi \in \text{cl}(\phi) \mid \epsilon \models \chi\} \tag{20}$$

$$F = \{\Psi \subseteq \text{cl}(\phi) \mid \phi \in \Psi\} \tag{21}$$

$$\delta(\Psi, \sigma) = \{\chi \in \text{cl}(\phi) \mid \Psi \overset{\sigma}{\to} \chi\} \tag{22}$$

where the relation $\Psi \xrightarrow{\sigma} \chi$, which intuitively means that if a string $w$ satisfies exactly the formulas in $\Psi$, then $w\sigma$ satisfies $\chi$, is defined as follows:

$$\Psi \xrightarrow{\sigma} \sigma' \text{ iff } \sigma = \sigma' \tag{23a}$$

$$\Psi \xrightarrow{\sigma} \chi_1 \wedge \chi_2 \text{ iff } \Psi \xrightarrow{\sigma} \chi_1 \text{ and } \Psi \xrightarrow{\sigma} \chi_2 \tag{23b}$$

$$\Psi \xrightarrow{\sigma} \neg\chi \text{ iff not } \Psi \xrightarrow{\sigma} \chi \tag{23c}$$

$$\Psi \xrightarrow{\sigma} \mathbf{Y}\chi \text{ iff } \chi \in \Psi \tag{23d}$$

$$\Psi \xrightarrow{\sigma} \mathbf{H}\chi \text{ iff } \mathbf{H}\chi \in \Psi \text{ and } \Psi \xrightarrow{\sigma} \chi \tag{23e}$$

$$\Psi \xrightarrow{\sigma} \chi_1 \mathbf{S} \chi_2 \text{ iff } (\chi_1 \mathbf{S} \chi_2 \in \Psi \text{ and } \Psi \xrightarrow{\sigma} \chi_1) \text{ or } \Psi \xrightarrow{\sigma} \chi_2. \tag{23f}$$

**Claim B.3.** *For any $w \in \Sigma^*$, if $\Psi = \{\chi \in \mathrm{cl}(\phi) \mid w \models \chi\}$, then $\Psi \xrightarrow{\sigma} \chi \iff w\sigma \models \chi$.*

*Proof.* By induction on the structure of $\chi$. Note by definition that $\chi \in \Psi \iff w \models \chi$.

$$\Psi \xrightarrow{\sigma} \sigma' \overset{(23a)}{\iff} \sigma = \sigma'$$
$$\overset{(11d)}{\iff} w\sigma \models \sigma'.$$

$$\Psi \xrightarrow{\sigma} \chi_1 \wedge \chi_2 \overset{(23b)}{\iff} \Psi \xrightarrow{\sigma} \chi_1 \text{ and } \Psi \xrightarrow{\sigma} \chi_2$$
$$\overset{\text{ind. hyp.}}{\iff} w\sigma \models \chi_1 \text{ and } w\sigma \models \chi_2$$
$$\overset{(11b)}{\iff} w\sigma \models \chi_1 \wedge \chi_2.$$

$$\Psi \xrightarrow{\sigma} \neg\chi \overset{(23c)}{\iff} \text{not } \Psi \xrightarrow{\sigma} \chi$$
$$\overset{\text{ind. hyp.}}{\iff} \text{not } w\sigma \models \chi$$
$$\overset{(11a)}{\iff} w\sigma \models \neg\chi.$$

$$\Psi \xrightarrow{\sigma} \mathbf{Y}\chi \overset{(23d)}{\iff} \chi \in \Psi$$
$$\iff w \models \chi$$
$$\overset{(11e)}{\iff} w\sigma \models \mathbf{Y}\chi.$$

$$\Psi \xrightarrow{\sigma} \mathbf{H}\chi \overset{(23e)}{\iff} \mathbf{H}\chi \in \Psi \text{ and } \Psi \xrightarrow{\sigma} \chi$$
$$\overset{\text{ind. hyp.}}{\iff} w \models \mathbf{H}\chi \text{ and } w\sigma \models \chi$$
$$\overset{(11f)}{\iff} w\sigma \models \mathbf{H}.$$

$$\Psi \xrightarrow{\sigma} \chi_1 \mathbf{S} \chi_2 \overset{(23f)}{\iff} (\chi_1 \mathbf{S} \chi_2 \in \Psi \text{ and } \Psi \xrightarrow{\sigma} \chi_1) \text{ or } \Psi \xrightarrow{\sigma} \chi_2$$
$$\overset{\text{ind. hyp.}}{\iff} (w \models \chi_1 \mathbf{S} \chi_2 \text{ and } w\sigma \models \chi_1) \text{ or } w\sigma \models \chi_2$$
$$\overset{(11g)}{\iff} w\sigma \models \chi_1 \mathbf{S} \chi_2. \qquad \square$$

**Claim B.4.** *For any $w$, $\delta(\iota, w) = \{\chi \in \mathrm{cl}(\phi) \mid w \models \chi\}$.*

*Proof.* By induction on the length of $w$.

Base case: $\delta(\iota, \epsilon) = \iota = \{\chi \mid \epsilon \models \chi\}$.

Inductive step: Assume that $\delta(\iota, w) = \{\chi \mid w \models \chi\} = \Psi$. Then

$$\delta(\iota, w) = \delta(\delta(\iota, w), \sigma)$$
$$= \delta(\Psi, \sigma)$$
$$= \{\chi \mid \Psi \xrightarrow{\sigma} \chi\}$$
$$= \{\chi \mid w\sigma \models \chi\}. \qquad \square$$

**Claim B.5.** *$M_\phi$ defines the same language as $\phi$.*

*Proof.* $\delta(\iota, w) \in F$ if and only if $\phi \in \{\chi \mid w \models \chi\}$ if and only if $w \models \phi$. $\qquad \square$

Then make every co-accessible state (every state that has a path to an accept state) into an accept state. Call this new DFA $M'_\phi$ with accept states $F'$. This DFA recognizes the prefix language of $M_\phi$. Finally, construct the formula

$$\mathrm{prefix}(\phi) = \bigvee_{\Psi \in F'} \left( \bigwedge_{\chi \in \Psi} \chi \wedge \bigwedge_{\chi \in \mathrm{cl}(\phi) \setminus \Psi} \neg\chi \right).$$

Note that prefix never translates a temporal operator into another temporal operator, so it translates formulas of $\mathsf{TL}[\mathcal{O}]$ into formulas of $\mathsf{TL}[\mathcal{O}]$ for any $\mathcal{O}$.

**Claim B.6.** *The formula* $\mathrm{prefix}(\phi)$ *defines the same language as* $M'_\phi$.

*Proof.* Since we only changed non-accept states to accept states, Clm. B.4 still applies to $M'_\phi$ and $\phi$.

$$
\begin{aligned}
\boldsymbol{w} \in \mathcal{L}(M'_\phi) &\iff \delta(\iota, \boldsymbol{w}) \in F' \\
&\iff \{\chi \in \mathrm{cl}(\phi) \mid \boldsymbol{w} \models \chi\} \in F' \qquad \text{Clm. B.4} \\
&\iff \text{for some } \Psi \in F', \chi \in \Psi \text{ iff } \boldsymbol{w} \models \chi \\
&\iff \text{for some } \Psi \in F', \boldsymbol{w} \models \bigwedge_{\chi \in \Psi} \chi \wedge \bigwedge_{\chi \in \mathrm{cl}(\phi) \setminus \Psi} \neg \chi \\
&\iff \boldsymbol{w} \models \bigvee_{\Psi \in F'} \left( \bigwedge_{\chi \in \Psi} \chi \wedge \bigwedge_{\chi \in \mathrm{cl}(\phi) \setminus \Psi} \neg \chi \right).
\end{aligned}
$$

This completes the proof of Lem. B.2. $\qquad \square$

### B.3 RELATIONSHIP BETWEEN CLASSIFIERS AND AUTOREGRESSORS

**Theorem 5.6.** *For any set of operators* $\mathcal{O} \subseteq \{\mathbf{Y}, \mathbf{H}, \mathbf{S}\}$:

(a) *For any nonempty language* $L$ *defined by a Boolean-weighted* $\mathsf{TL}[\mathcal{O}]$ *classifier, there exists a Boolean-weighted* $\mathsf{TL}[\mathcal{O}]$ *autoregressor defining the same language* $L$.

(b) *For any language* $L$ *defined by a Boolean-weighted* $\mathsf{TL}[\mathcal{O}]$ *autoregressor, there exists a Boolean-weighted* $\mathsf{TL}[\mathcal{O} \cup \{\mathbf{Y}, \mathbf{H}\}]$ *classifier defining the same language* $L$.

*Proof.* (a) A Boolean-weighted $\mathsf{TL}[\mathcal{O}]$ classifier is defined by a tuple of formulas $\Phi = (\phi_1, \ldots, \phi_m)$ inducing a state encoder and an output function $c \colon \mathbb{B}^m \to \mathbb{B}$. We may think of $c$ as a Boolean combination of its arguments, and substitute the $\phi_i$ into it to obtain a single formula $\phi = c(\phi_1, \ldots, \phi_m)$. Then define a new trivial output function $c'(h) = h$, so that the classifier $(\phi, c')$ defines the same language as $(\Phi, c)$.

Define

$$
\begin{aligned}
\phi_\sigma &= \mathrm{next}_\sigma(\mathrm{prefix}(\phi)) \qquad \text{for } \sigma \in \Sigma & (24) \\
\phi_{\mathrm{EOS}} &= \phi. & (25)
\end{aligned}
$$

Then the tuple of formulas $\Phi' = (\phi_\sigma)_{\sigma \in \Sigma \cup \{\mathrm{EOS}\}}$ defines a state encoder. We define the autoregressive output function

$$
r \colon \mathbb{B}^{|\Sigma|+1} \to \mathbb{B}^{|\Sigma|+1} \tag{26}
$$

$$
r(\mathbf{h})(\sigma) = \begin{cases} h_\sigma & \text{if any entry of } \mathbf{h} \text{ is true} \\ \top & \text{if all entries of } \mathbf{h} \text{ are false.} \end{cases} \tag{27}
$$

A vector $\mathbf{h}$ whose entries are all false is unreachable, so it does not matter what we set $r(\mathbf{h})$ to, but it must satisfy $\bigoplus_\sigma r(\mathbf{h})(\sigma) = \mathbf{1}$ (Eq. (7)), that is, the entries of $r(\mathbf{h})$ cannot all be false.

The autoregressor $A = (\Phi', r)$ defines $L$, because for any $\boldsymbol{w} \in L$ with length $n$, we have

$$\Pr_A(\boldsymbol{w}) = \bigotimes_{i=1}^{n} \Pr_A(w_i \mid \boldsymbol{w}_{<i}) \otimes \Pr_A(\text{EOS} \mid \boldsymbol{w}) \tag{28}$$

$$= \bigwedge_{i=1}^{n} \mathbb{I}\{\boldsymbol{w}_{<i} \models \text{next}_{w_i}(\text{prefix}(\phi))\} \wedge \mathbb{I}\{\boldsymbol{w} \models \phi\} \tag{29}$$

$$= \bigwedge_{i=1}^{n} \mathbb{I}\{\boldsymbol{w}_{\leq i} \models \text{prefix}(\phi)\} \wedge \mathbb{I}\{\boldsymbol{w} \models \phi\} \tag{30}$$

$$= \bigwedge_{i=1}^{n} \mathbb{I}\{\boldsymbol{w}_{\leq i}\boldsymbol{v} \models \phi \text{ for some } \boldsymbol{v}\} \wedge \mathbb{I}\{\boldsymbol{w} \models \phi\} \tag{31}$$

$$= \top. \tag{32}$$

On the other hand, for any $\boldsymbol{w} \notin L$, let $k$ be the greatest integer such that $\boldsymbol{w}_{<k}\boldsymbol{v} \in L$ for some $\boldsymbol{v}$. (We know that $k$ exists because $L$ is nonempty by assumption.) Then we have that $\boldsymbol{w}_{<k} \not\models \text{next}_{w_k}(\text{prefix}(\phi))$, but $\boldsymbol{w}_{<k} \models \text{next}_{v_1}(\text{prefix}(\phi))$. So $\Pr_A(w_k \mid \boldsymbol{w}_{<k}) = \bot$, and therefore $\Pr_A(\boldsymbol{w}) = \bot$.

It remains to verify that $A$ satisfies Eq. (6). For any $\boldsymbol{u} \in \Sigma^*$, we want to show that

$$\bigoplus_{\boldsymbol{v}} \Pr_A(\boldsymbol{v} \mid \boldsymbol{u}) = \bigoplus_{\boldsymbol{v}} \left( \bigotimes_{i=1}^{|\boldsymbol{v}|} \Pr_A(v_i \mid \boldsymbol{u}\boldsymbol{v}_{<i}) \otimes \Pr_A(\text{EOS} \mid \boldsymbol{u}\boldsymbol{v}) \right) = \mathbf{1}. \tag{33}$$

In the Boolean semiring, it suffices to show that at least one term of this summation is true.

If there is a $\boldsymbol{v}$ such that $\boldsymbol{u}\boldsymbol{v} \in L$, then the corresponding term of the summation is

$$\left( \bigwedge_{i=1}^{|\boldsymbol{u}\boldsymbol{v}|} \mathbb{I}\{\boldsymbol{u}\boldsymbol{v}_{<i} \models \text{next}_{v_i}(\text{prefix}(\phi))\} \right) \wedge \mathbb{I}\{\boldsymbol{u}\boldsymbol{v} \models \phi\} \tag{34}$$

$$= \left( \bigwedge_{i=1}^{|\boldsymbol{u}\boldsymbol{v}|} \mathbb{I}\{\boldsymbol{u}\boldsymbol{v}_{\leq i} \models \text{prefix}(\phi)\} \right) \wedge \mathbb{I}\{\boldsymbol{u}\boldsymbol{v} \models \phi\} \tag{35}$$

$$= \left( \bigwedge_{i=1}^{|\boldsymbol{u}\boldsymbol{v}|} \mathbb{I}\{\boldsymbol{u}\boldsymbol{v}_{\leq i}\boldsymbol{w} \models \phi \text{ for some } \boldsymbol{w}\} \right) \wedge \mathbb{I}\{\boldsymbol{u}\boldsymbol{v} \models \phi\} \tag{36}$$

$$= \top. \tag{37}$$

If there is no such $\boldsymbol{v}$, then for all $\sigma$, we have $\boldsymbol{u}\sigma \not\models \text{prefix}(\phi)$, so $\boldsymbol{u} \not\models \text{next}_\sigma(\text{prefix}(\phi))$; moreover, $\boldsymbol{u} \not\models \phi$. Let $\mathbf{h} = \Phi'(\boldsymbol{u})_{|\boldsymbol{u}|}$ be the state after reading $\boldsymbol{u}$. All entries of $\mathbf{h}$ are false, which makes (for example) $a(\mathbf{h})(\text{EOS})$ true, so the $\boldsymbol{v} = \epsilon$ term of the summation in Eq. (33) is true.

(b) Let $A = (\Phi, r)$ be a $\mathsf{TL}[\mathcal{O}]$ autoregressor, where $\Phi = (\phi_i)_{i=1}^{m}$.

Define

$$\phi_{\mathbf{h}} = \bigwedge_{i=1}^{m} (\phi_i \leftrightarrow h_i) \qquad\qquad \text{for } \mathbf{h} \in \mathbb{B}^m$$

$$\phi_\sigma = \bigwedge_{\mathbf{h} \in \mathbb{B}^m} (\phi_{\mathbf{h}} \leftrightarrow r(\mathbf{h})(\sigma)) \qquad\qquad \text{for } \sigma \text{ in } \Sigma \cup \{\text{EOS}\}.$$

Intuitively, $\phi_h$ is true whenever the state encoder is in state $\mathbf{h}$, and $\phi_\sigma$ is true whenever the state encoder is in a state in which $r$ predicts $\sigma$.

Then, define

$$\phi' = \mathbf{H} \left( \bigvee_{\sigma \in \Sigma \cup \{\text{EOS}\}} (\mathbf{Y}\phi_\sigma) \wedge \sigma \right)$$

and let $c(h) = h$, so that the classifier $C = (\phi', c)$ defines the same language as $A$.

$\square$

## B.4 PROOF OF PROP. 5.7

**Proposition 5.7.** *(a) There does not exist a transformation* $\text{prefix}'$ *such that* $\text{prefix}'(\phi)$ *is constructible in polynomial time (in* $|\phi|$*) and satisfies Eq. (19) for every formula* $\phi$ *in* $\text{TL}[\mathbf{H}, \mathbf{Y}]$*, unless* $\mathsf{P} = \mathsf{PSPACE}$.
*(b) Similarly for* $\text{TL}[\mathbf{H}]$*, unless* $\mathsf{P} = \mathsf{NP}$.
*(c) Similarly for* $\text{TL}[\mathbf{Y}]$*, unless* $\mathsf{P} = \mathsf{NP}$.

*Proof.* (a) Suppose that $\text{prefix}'$ exists. For any formula $\phi$ of $\text{TL}[\mathbf{H}, \mathbf{Y}]$, we can test whether $\phi$ is satisfiable by constructing $\text{prefix}'(\phi)$ in polynomial time (by assumption) and then testing whether $\epsilon \models \text{prefix}'(\phi)$, which can also be done in polynomial time, as shown by Fionda and Greco (2016, Thm. 8). (They assume formulas in negation normal form, but it is easy to generalize their result to formulas not in negation normal form.) But satisfiability in $\text{TL}[\mathbf{H}, \mathbf{Y}]$ is PSPACE-complete (Giacomo and Vardi, 2013; Fionda and Greco, 2016), so this would imply $\mathsf{P} = \mathsf{PSPACE}$.

(b) Similarly, satisfiability in $\text{TL}[\mathbf{H}]$ is NP-complete (Fionda and Greco, 2016), so the existence of a polynomial-time $\text{prefix}'$ would imply $\mathsf{P} = \mathsf{NP}$.

(c) Same as the previous case.

$\square$

## C NONDETERMINISTIC FINITE AUTOMATA

We give a single definition of weighted NFAs instead of factoring them into unweighted NFAs and autoregressive output functions.

**Definition C.1** (Weighted Nondeterministic Finite Automaton). *A **weighted nondeterministic finite automaton** is a tuple* $M = (\Sigma, Q, \delta, \iota, \omega)$*, where*

- $\Sigma$ *is an alphabet*

- $Q$ *is a finite set of **states***

- $\delta \colon Q \times \Sigma \times Q \to \mathbb{K}$ *is a **transition function***

- $\iota \in Q$ *is the **initial** state*

- $\omega \colon Q \to \mathbb{K}$ *is the **accept function**.*

*We extend $\delta$ to $\delta^* \colon Q \times \Sigma^* \times Q \to \mathbb{K}$:*

$$\delta^*(q, \epsilon, q) = \mathbf{1}$$
$$\delta^*(q, \epsilon, q') = \mathbf{0} \qquad q \neq q'$$
$$\delta^*(q_1, \sigma \boldsymbol{w}, q_2) = \bigoplus_{q \in Q} \delta(q_1, \sigma, q) \otimes \delta^*(q, \boldsymbol{w}, q_2).$$

*Then $M$ accepts $\boldsymbol{w}$ with weight $k$ iff*

$$k = \bigoplus_{q_2 \in Q} \delta^*(\iota, \boldsymbol{w}, q_2) \otimes \omega(q_2).$$

**Definition C.2.** *We say that an NFA with transition function $\delta$ is **counter-free** if there exists some $k$ such that for all states $q_1, q_2$ and all strings $\boldsymbol{w}$, we have $\delta^*(q_1, \boldsymbol{w}^k, q_2) = \delta^*(q_1, \boldsymbol{w}^{k+1}, q_2)$.*

The equivalence between counter-free NFAs and DFAs is well-known (e.g., McNaughton and Papert, 1971, Chapter 5, Exercise 15), but we spell out the proof here using our definitions.

**Proposition C.1.** *With Boolean weights, counter-free NFAs and counter-free DFAs are equivalent.*

*Proof.* First, any Boolean-weighted counter-free NFA $M = (\Sigma, Q, \delta, \iota, \omega)$ can be converted into an equivalent counter-free DFA $M' = (\Sigma, Q', \delta', \iota')$ together with a classifier output function $\omega' \colon Q' \to \mathbb{B}$. This is done by the standard construction (Sipser, 2013), letting $Q' = 2^Q$ and defining $\delta'^*(\bar{q}, \boldsymbol{w}) = \{r \mid q \in \bar{q}, \delta^*(q, \boldsymbol{w}, r) = \mathbf{1}\}$, $\iota' = \{\iota\}$, and $\omega'(\bar{q}) = \bigoplus_{q \in \bar{q}} \omega(q)$. Then $\delta'^*(\bar{q}, \boldsymbol{w}^k) = \{r \mid \exists q \in \bar{q}.\delta^*(q, \boldsymbol{w}^k, r) = \mathbf{1}\} = \{r \mid \exists q \in \bar{q}.\delta^*(q, \boldsymbol{w}^{k+1}, r) = \mathbf{1}\} = \delta'^*(\bar{q}, \boldsymbol{w}^{k+1})$. Thus, $M'$ is also counter-free. The other direction is trivial. $\qquad\square$

A weighted NFA is determinizable if every pair of states which are *siblings* (can be reached by the same string) are also *twins* (all cycles by the same string have the same weight) (Mohri, 1997). The automaton in Fig. 2c is counter-free, but not determinizable, because $q_1$ and $q_2$ are siblings (both reachable by $a$) but not twins (the $a$-labeled cycles $q_1 \xrightarrow{a/\frac{1}{2}} q_1$ and $q_2 \xrightarrow{a/\frac{3}{4}} q_2$ on the two states have different weights).

## D  INEXPRESSIBILITY OF $(aab)^*$

**Proposition 5.9.** *The language $(aab)^*$ is not definable by any $\mathsf{TL}[\mathbf{H}]$ classifier or autoregressor.*

We actually prove a slightly stronger statement. Define the **Y-depth** of a formula $\phi$ to be the number of nested $\mathbf{Y}$ operators in $\phi$. Then we will prove that $(aab)^*$ is not definable by any formula of $\mathsf{TL}[\mathbf{H}, \mathbf{Y}]$ with $\mathbf{Y}$-depth 1. Since the conversion from an autoregressor to a classifier (Thm. 5.6(b)) adds a single $\mathbf{Y}$, we will conclude that $(aab)^*$ is not definable by any $\mathsf{TL}[\mathbf{H}]$ autoregressor.

**Lemma D.1.** *For any language $L$ over $\Sigma$, define* $\mathrm{Bigram}(L) = \{(\mathrm{BOS}, w_1) \cdot (w_1, w_2) \cdot (w_2, w_3) \cdots (w_{n-1}, w_n) \cdot (w_n, \mathrm{EOS}) \mid w \in L\}$. *If $\phi$ is a formula of $\mathsf{TL}[\mathbf{H}, \mathbf{Y}]$ with $\mathbf{Y}$-depth 1, then there is a formula $\mathrm{noy}(\phi)$ of $\mathsf{TL}[\mathbf{H}]$ over $(\Sigma \cup \{\mathrm{BOS}\}) \times (\Sigma \cup \{\mathrm{EOS}\})$ such that $\mathcal{L}(\mathrm{noy}(\phi)) \cap \mathrm{Bigram}(\Sigma^*) = \mathrm{Bigram}(\mathcal{L}(\phi))$.*

*Proof.* Define the transformation noy, which pushes $\mathbf{Y}$ down to the atomic formulas, then modifies the atomic formulas to operate on bigrams.

$$\mathrm{noy}(\neg\psi) = \neg\mathrm{noy}(\psi) \qquad\qquad \mathrm{noy}(\mathbf{Y}(\neg\psi)) = \neg\mathrm{noy}(\mathbf{Y}\psi)$$
$$\mathrm{noy}(\psi_1 \wedge \psi_2) = \mathrm{noy}(\psi_1) \wedge \mathrm{noy}(\psi_2) \qquad \mathrm{noy}(\mathbf{Y}(\psi_1 \wedge \psi_2)) = \mathrm{noy}(\mathbf{Y}\psi_1) \wedge \mathrm{noy}(\mathbf{Y}\psi_2)$$
$$\mathrm{noy}(\mathbf{H}\psi) = \mathbf{H}(\mathrm{noy}(\psi)) \qquad\qquad \mathrm{noy}(\mathbf{Y}(\mathbf{H}\psi)) = \mathbf{H}(\mathrm{noy}(\mathbf{Y}\psi))$$
$$\mathrm{noy}(\sigma) = \bigvee_{\sigma' \in \Sigma \cup \{\mathrm{BOS}\}} (\sigma', \sigma) \qquad\qquad \mathrm{noy}(\mathbf{Y}\sigma) = \bigvee_{\sigma' \in \Sigma \cup \{\mathrm{EOS}\}} (\sigma, \sigma')$$
$$\mathrm{noy}(\mathrm{BOS}) = \mathrm{BOS} \qquad\qquad \mathrm{noy}(\mathbf{Y}\,\mathrm{BOS}) = \bigvee_{\sigma' \in \Sigma \cup \{\mathrm{EOS}\}} (\mathrm{BOS}, \sigma'). \qquad\square$$

Then, to prove that $(aab)^*$ is not definable in $\mathsf{TL}[\mathbf{H}, \mathbf{Y}]$ with $\mathbf{Y}$-depth 1, suppose it is definable by $\phi$. By Lem. D.1, there is a formula $\mathrm{noy}(\phi)$ of $\mathsf{TL}[\mathbf{H}]$ such that

$$(\mathrm{BOS}, a) \cdot (a, a) \cdot (a, b) \cdot (b, \mathrm{EOS}) \in \mathcal{L}(\mathrm{noy}(\phi)) \cap \mathrm{Bigram}(\Sigma^*).$$

But $\mathcal{L}(\mathrm{noy}(\phi))$ must be stutter-invariant (Peled and Wilke, 1997), so we also have

$$(\mathrm{BOS}, a) \cdot (a, a) \cdot (a, a) \cdot (a, b) \cdot (b, \mathrm{EOS}) \in \mathcal{L}(\mathrm{noy}(\phi)) \cap \mathrm{Bigram}(\Sigma^*).$$

But this is not in $\mathrm{Bigram}((aab)^*)$, which is a contradiction.

