# OpenReview forum: "Probability Distributions Computed by Autoregressive Transformers"
_ICLR.cc/2026/Conference — ICLR 2026 Poster_

### Official Review · Reviewer_hwiM · 2025-10-30

**Soundness:** 3
**Presentation:** 2
**Contribution:** 3
**Rating:** 4
**Confidence:** 2

**Summary:**

The paper advances the theoretical study of the expressivity of transformer language models beyond Boolean recognizers, by characterizing their expressivity as probabilistic, autoregressors. The paper focuses on Unique-Hard Attention Transformers (UHATs) and connects them with Linear Temporal Logic (LTL) and counter-free Deterministic Finite Automata (cfDFA) paradigms. Overall, the paper proposes several expressivity results for these paradigms, characterizing when different modeling choices (classifier vs. autoregressive paradigms, Boolean vs. real semirings) do or do not change the set of weighted languages a model can represent.

**Strengths:**

1. The theoretical results are meaningful and timely, which advance the theoretical understanding of the expressivity of transformer-based models.


2. In my view, the main strength of the paper is its focus on the probabilistic autoregressive regime, which is more aligned with how language models are used in practice, and thus represents a more meaningful case of study in comparison to previous works.


3. The paper characterizes equivalences in terms of language recognition of UHATs with known classic models (LTL, DFA…). Furthermore, describe different levels of expressivity under different scenarios or configurations.

**Weaknesses:**

1. These results hold for unique-hard attention with no position embeddings. This diverges from common configurations of language models (which include soft attention and positional encodings). Furthermore, previous works such as (Li and Cotterell, 2025) focus on soft attention, resulting in a less-idealized setup (closer to real-world transformers). I believe the paper would significantly benefit from a discussion on which results will hold or not (and why) under these assumptions.


2. Some results are somewhat incremental, based mostly on the results of (Yang et al., 2024) or (Li and Cotterell, 2025). While I value the theoretical results, I believe that this limits the contribution of this particular work.


3. While the paper adequately describes its goals in Section 1, its novelty and the main contributions are unclear or hard to follow. I would recommend the authors to reinforce this part, in order to emphasize which are the novel results / extensions of this particular work (with respect to previous related works), and the relevance of its contributions.


4. The paper lacks a discussion of potential future research directions building on these findings.

**Questions:**

1. Which results extend to soft attention? Similarly, which are the implications of positional embeddings? I would recommend a discussion on these matters.


2. Related to my previous point, recent works show that hard attention can be simulated using soft attention through temperature scaling, e.g., see (Yang et al, 2024). Based on this, will some of your results hold under soft attention? Can these findings bridge the results from this paper to soft-attention regimes?

3. The Related Work section seems shallow in its current state. This is mostly because most of the closest works are described in Section 1. I would encourage the authors to reorganize Sections 1 and 2.


4. Furthermore, some recent works are missing in the literature review (e.g., see the list below). Some of these missing works address similar scenarios or goals. For example, (Yang et al, 2024) also examines several subclasses of languages recognized by hard-attention transformers, which can be defined in variants of linear temporal logic. Please clarify the similarities or differences with these works.

- Yang, A., Strobl, L., Chiang, D., & Angluin, D. (2024). Simulating hard attention using soft attention. arXiv preprint arXiv:2412.09925.


- Hao, Y., Angluin, D., & Frank, R. (2022). Formal language recognition by hard attention transformers: Perspectives from circuit complexity. Transactions of the Association for Computational Linguistics, 10, 800-810.

- Barceló, P., Kozachinskiy, A., Lin, A. W., & Podolskii, V. (2023). Logical languages accepted by transformer encoders with hard attention. ICLR 2024.




5. Please define the acronym UHAT as “unique-hard attention transformers” in Page 1, for the sake of clarity to non-familiar readers.


6. Relevant results from these works focus on counter-free automata. I wonder whether this limits impact on certain cases that exhibit periodicity. A brief discussion of what is left outside the counter-free setting and whether partial extensions or approximations are possible would be helpful.


7. While the paper focuses on expressivity results, can something be said about “learnability” (i.e., sample complexity, efficiency…)?


8. Sections 6.2 and 6.3 lack explicit UHAT results. I believe that the paper might benefit from a more clear discussion on how those results contribute to the study of UHAT expressivity.


9. Please include a forward-looking discussion of open problems and next steps.

---

> ### Author Response · Authors · 2025-11-21
> **Rebuttal by Authors, Part 1**
>
> # Response to Reviewer hwiM
>
> Thank you very much for your review.
>
> > These results hold for unique-hard attention with no position embeddings. This diverges from common configurations of language models (which include soft attention and positional encodings). Furthermore, previous works such as (Li and Cotterell, 2025) focus on soft attention, resulting in a less-idealized setup (closer to real-world transformers). I believe the paper would significantly benefit from a discussion on which results will hold or not (and why) under these assumptions.
>
>
> See our response on this in the first question!
>
> > Some results are somewhat incremental, based mostly on the results of (Yang et al., 2024) or (Li and Cotterell, 2025). While I value the theoretical results, I believe that this limits the contribution of this particular work.
>
> If our theoretical results are seen as incremental, this is only because our novel theoretical framework enabled previous work to be adapted towards the autoregressive setting. We would like to highlight that no previous work considered what probability distributions transformer language models could express (we will improve the introduction to make this contribution more clear). We had to formalize of autoregression in a manner that is compatible both with the Boolean and probability semirings. The fact that our results appear straightforward in this framework is just evidence that we have uncovered the correct formal notion of autoregression, which we believe is a significant contribution. See lines 29-33 for an updated discussion on our contributions.
>
> > While the paper adequately describes its goals in Section 1, its novelty and the main contributions are unclear or hard to follow. I would recommend the authors to reinforce this part, in order to emphasize which are the novel results / extensions of this particular work (with respect to previous related works), and the relevance of its contributions.
>
> We agree, and have updated the introduction to clarify our contributions with respect to previous and related works.
>
> > The paper lacks a discussion of potential future research directions building on these findings.
>
> We have added discussion of promising future research directions in Section 7.
>
>
> ## Questions
>
> > Which results extend to soft attention? Similarly, which are the implications of positional embeddings? I would recommend a discussion on these matters.
>
> Regarding soft attention note that our paper covers softmax attention with two different notions of fixed-precision, but extension to to softmax attention (e.g., with log precision) is an open question. There aren't even yet (to our knowledge) any exact characterizations of transformer classifiers. Given such a characterization, we think our theoretical framework could be adapted to analyze the differences between the classification and autoregression settings. Presumably, the notion of "state" would have to be extended from a finite set to a set of size $O(\log n)$.
>
> There are existing upper and lower bounds on log-precision SMATs, and we leave adaptation of these bounds to autoregressors for future work. In particular, log-precision transformers as classifiers are in $\mathsf{TC}^0$ (Merrill & Sabharwal, 2023), and we conjecture that log-precision transformers as autoregressors remain in $\mathsf{TC}^0$.
>
> Regarding positional embeddings, our framework can easily be used to analyze different kinds of positional embeddings by leveraging their connections with different operators in temporal logic. For instance, ALiBi is connected with the $\mathbf{Y}$ operator, and sinusoidal/rotary postional embeddings are connected with modular predicates (Yang et al, 2025). We save a detailed analysis for future work, but suspect many conclusions will be similar to those currently in the paper.
>
> We have added a discussion on these matters in Section 7.

---

> > ### Author Response · Authors · 2025-11-21
> > **Rebuttal by Authors, Part 2**
> >
> > > Related to my previous point, recent works show that hard attention can be simulated using soft attention through temperature scaling, e.g., see (Yang et al, 2024). Based on this, will some of your results hold under soft attention? Can these findings bridge the results from this paper to soft-attention regimes?
> >
> > This is an interesting question. These results touch on lower apply to lower bounds on expressivity. In particuclar, the results of (Yang et al, 2024) combined with our Theorem 6.4 imply that any language recognized by a unique-hard attention transformer classifer or autoregressor can be recognized by a soft attention classifier.
> >
> > It is open whether the upper bounds will remain the same. We conjecture with moderate confidence that existing upper bounds will not be affected. More precisely, log precision transformer as classifiers are upper bounded by $\mathsf{TC}^0$ (Merrill & Sabharwal, 2023), and we conjecture autoregressive log precision transformers remain in $\mathsf{TC}^0$.
> >
> >
> > >The Related Work section seems shallow in its current state. This is mostly because most of the closest works are described in Section 1. I would encourage the authors to reorganize Sections 1 and 2.
> >
> > We have kept the essential background references in the introduction, and expanded the related works section to include other references suggested by reviewers, as well as moving it to the end of the paper. Thank you for the suggestions!
> >
> > > Furthermore, some recent works are missing in the literature review (e.g., see the list below). Some of these missing works address similar scenarios or goals. For example, (Yang et al, 2024) also examines several subclasses of languages recognized by hard-attention transformers, which can be defined in variants of linear temporal logic. Please clarify the similarities or differences with these works.
> >
> > We agree these references are relevant, and had intended to subsume some of them by the reference to Strobl et al (2024)'s survey. We added notes to some of these in the Related Works section, especially to Yang et al, whose work further demonstrates the effectiveness of hard attention.
> >
> > >> Yang, A., Strobl, L., Chiang, D., & Angluin, D. (2024). Simulating hard attention using soft attention. arXiv preprint arXiv:2412.09925.
> >
> > >> Hao, Y., Angluin, D., & Frank, R. (2022). Formal language recognition by hard attention transformers: Perspectives from circuit complexity. Transactions of the Association for Computational Linguistics, 10, 800-810.
> >
> > >>Barceló, P., Kozachinskiy, A., Lin, A. W., & Podolskii, V. (2023). Logical languages accepted by transformer encoders with hard attention. ICLR 2024.
> >
> > >Please define the acronym UHAT as “unique-hard attention transformers” in Page 1, for the sake of clarity to non-familiar readers.
> >
> > Thanks for catching this mistake, which we have corrected.
> >
> > >Relevant results from these works focus on counter-free automata. I wonder whether this limits impact on certain cases that exhibit periodicity. A brief discussion of what is left outside the counter-free setting and whether partial extensions or approximations are possible would be helpful.
> >
> > Our framework can be easily extended to handle some cases of periodicity which transformers can handle in practice. We could add sinusoidal positional encodings to the transformers, which corresponds to adding modular predicates to the logic. We suspect this will result an equivalence with quasi-counter-free automata.
> >
> > >While the paper focuses on expressivity results, can something be said about “learnability” (i.e., sample complexity, efficiency…)?
> >
> > This is a very interesting question which we leave for future work. Our focus was the expressivity of the model, which builds a foundation for future work on learnability.
> >
> > >Sections 6.2 and 6.3 lack explicit UHAT results. I believe that the paper might benefit from a more clear discussion on how those results contribute to the study of UHAT expressivity.
> >
> > We stated the results in terms of abstract state encoders for generality and ease of writing, though these results apply equally to the classes of hard-attention transformers equivalent to these logics. Figure 1 makes these connections explicit.
> >
> >
> > >Please include a forward-looking discussion of open problems and next steps.
> >
> > We have added a future work section with several promising extensions and directions to our work.
> >
> > Merrill & Sabharwal, 2023: A Logic for Expressing Log-Precision Transformers. NeurIPS. arXiv:2210.02671
> >
> > Strobl et al. 2024: What Formal Languages Can Transformers Express? A Survey. TACL. arXiv: 2311.00208
> >
> > Yang et al, 2024: Simulating Hard Attention Using Soft Attention. arXiv:2412.09925
> >
> > Yang et al, 2025: Knee-Deep in C-RASP: A Transformer Depth Hierarchy. NeurIPS. arXiv:2506.16055

---

### Official Review · Reviewer_DAYT · 2025-10-30

**Soundness:** 3
**Presentation:** 3
**Contribution:** 3
**Rating:** 6
**Confidence:** 3

**Summary:**

This paper studies the expressive power of transformers as probabilistic autoregressive models in the context of formal language theory. Existing theoretical work mostly focuses on transformers as language recognizers (Boolean setting). Here, the authors analyze the probability distributions computed by hard-attention transformers when used as language models, providing equivalence results and separation results between Boolean classifiers, probabilistic classifiers, and probabilistic autoregressors. The study also connects these models with temporal logics and weighted automata, clarifying how expressivity shifts across these settings.
The paper is rigorous, well-structured, and contributes to closing a gap in theoretical understanding of transformer expressivity in autoregressive scenarios.

**Strengths:**

* Tightly written theoretical work with clear formal contributions.
* Addresses a meaningful gap in theory: expressive power of transformers as generative language models.
* Strong formal rigor, with proofs and precise definitions throughout the paper.

**Weaknesses:**

* **Purely theoretical:** While the theoretical contributions are solid, there are no experimental results nor concrete applied examples to illustrate relevance for real-world transformer LMs. Given the conference venue, this limits perceived impact.
* **Accessibility:** The paper assumes familiarity with temporal logics, weighted automata, and semirings. This is appropriate for a specialized logic or theoretical CS audience, but is demanding for the general machine-learning readership at ICLR.

Minor Comments:
* There are small repetitions in early sections (e.g., “we then” in lines ~77-82, “language” in lines ~90-93) that make the text slightly repetitive.
* The notion of “state” is only clarified around Section 4-5. Since “state encoder” is central, I suggest introducing more intuitively what a “state” represents earlier in the introduction.
* Typo in Section 6.1: $\tau_1$ should be $\tau_2$.

**Questions:**

I do not have any particular questions for the authors.

---

> ### Author Response · Authors · 2025-11-21
> **Rebuttal by Authors**
>
> # Response to Reviewer DAYT
>
> Thank you very much for your review.
>
> > While the theoretical contributions are solid, there are no experimental results nor concrete applied examples to illustrate relevance for real-world transformer LMs. Given the conference venue, this limits perceived impact.
>
> We agree that an empirical validation, particularly one involving real-world transformer LMs, is a crucial next step. In fact, we have already run a set of preliminary experiments to investigate this, in both classification and language modeling settings.
>
> Our initial findings were mixed:
> 1. On one hand, we found that non-star-free languages are indeed learned more poorly, which supports our theoretical claims.
> 2. On the other hand, the empirical difference between languages definable and not definable in $\mathsf{TL}[\mathbf{P}]$ was not clear-cut, and these results were not conclusive.
>
> We concluded that these initial experiments are not yet comprehensive enough to include. The ambiguity in the $\mathsf{TL}[\mathbf{P}]$ results suggests that designing a conclusive experiment to isolate these specific properties in pracitcal LMs is a significant challenge in its own right and requires deeper investigation.
>
> Therefore, rather than including an incomplete empirical study that might detract from the main message, we have decided to maintain the paper's focus on its theoretical contributions.
>
> Regarding relevance for real-world transformer LMs:
>
> A wealth of research on transformers as classifiers (Strobl et al. 2024) has clarified the abilities and limitations of transformers on many algorithmic tasks, but it was previously unclear whether this extended to the autoregressive setting. Our results suggest that we can usually expect results on the expressivity classifiers to apply to autoregressors. We have tried to make this clearer in the revision. Some points that can be made are:
>
> - With softmax attention, autoregressive transformers should fail at recognizing the word problem for $S_5$, Boolean formula evaluation, and other algorithmic tasks difficult for softmax transformer classifiers.
> - With hard attention, autoregressive transformers should fail at recognizing $\mathsf{PARITY}$, $\mathsf{MAJORITY}$, $\mathsf{DYCK}$-$\mathsf{1}$, and other algorithmic tasks difficult for hard-attention transformer classifiers.
>
> Additionally, Section 6.3.3 makes a subtle point about practical settings; we clarified the separation between classification and autoregression in the experiments of Yang et al (2025), who found that with $k$ layers, a transformer classifier could recognize $k$ blocks of alternating symbols, but an autoregressor could recognize $k+2$.
>
> > Accessibility: The paper assumes familiarity with temporal logics, weighted automata, and semirings. This is appropriate for a specialized logic or theoretical CS audience, but is demanding for the general machine-learning readership at ICLR.
>
> We've provided self-contained definitions of all necessary concepts in temporal logics, weighted automata, and semirings, but these definitions are admittedly quite brief. We've provided a pointer to a more detailed introduction, and we've revised the paper to add pointers to introductions to temporal logic and weighted automata as well.
>
> >The notion of “state” is only clarified around Section 4-5. Since “state encoder” is central, I suggest introducing more intuitively what a “state” represents earlier in the introduction.
>
> Thank you for this suggestion. We now make a brief mention to the term "state encoder" in the introduction, and at the beginning of Section 5 we define it more properly.
>
> Strobl et al. 2024: What Formal Languages Can Transformers Express? A Survey. TACL. arXiv: 2311.00208
>
> Yang et al, 2025: Knee-Deep in C-RASP: A Transformer Depth Hierarchy. NeurIPS. arXiv:2506.16055

---

> > ### Comment · Reviewer_DAYT · 2025-11-26
> > **Thank you for the reply**
> >
> > Thank you for addressing my concerns. I agree with the authors that including empirical evaluation is out-of-scope and should not be addressed in this work, especially due to the page limit imposed by the conference.
> > For this reason, I still believe this work suits more in a top-tier journal rather than a conference, where the authors have more space to clearly analyze all the details and implications of their idea.
> > That said, the work sounds great, and I believe that a rating of 6 fits well.

---

### Official Review · Reviewer_A2Nv · 2025-10-31

**Soundness:** 3
**Presentation:** 3
**Contribution:** 3
**Rating:** 6
**Confidence:** 3

**Summary:**

This work studies the expressive capacity of Transformers when used as probabilistic language models rather than classifiers. To do so, the authors analyze models along 2 axes: 1) Classifiers vs Autoregressors 2) Boolean weights vs (positive) Real-valued weights. The authors show the following:
- UHATs, LTL and cfDFAs have equivalent state encoders (Thm 6.1) They use this to show that UHATs, LTLs and cfDFAs as classifiers (or autoregressors) define the same weighted language (Corollary 6.1)
- LTL classifiers can only output finitely many distinct weight values (Prop 6.1). This implies that LTL classifiers cannot express the language $(\frac{1}{2} a)^*$, which can be expressed by autoregressors (Corollary 6.2).
- LTL classifiers and autoregressors are equivalent (for the right subsets of operators) (Thm 6.2)
- In the case of autoregressors, cfNFAs are more expressive than cfDFAs (Prop 6.2)
- Boolean autoregressors are more expressive than classifiers when key operators are missing (Prop. 6.3)
- The expressivity gain of autoregression is limited; $(aab)^*$ is not definable by an LTL regressor (Prop 6.4 & Thm 6.3)

I think this is a good paper with solid theoretical contributions. However, I find the style in which it is written makes it hard to quickly extract the main insights and results. I have given a 6 and will increase my score to an 8 if the authors address points 3, 4 and 7 in the Questions/Comments section.

**Strengths:**

- Angle is novel and interesting. I agree with the authors that there is a lack of literature on the topic of language *modelling* with Transformers.
- This work improves our understanding of the interplay of expressive power between i) Boolean and Real-valued models and ii) classifier and autoregressive models. Given the equivalencies drawn by the authors, the theoretical results have deep implications about many families of models.
- The authors provide extremely rigorous proofs and reductions. From a technical perspective. The theoretical approach taken is non-trivial and is in itself a significant contribution to theoretical research. Although I am knowledgeable about the Transformer expressivity literature, I am not well-versed in temporal logic, thus I was not able to fully check all the proofs pertaining to this in detail.

**Weaknesses:**

- Although complete and precise in its writing style, I find the paper is very dense and not written in a way where key insights are easy to find/extract. See comments for actionable feedback.
- The paper has no experimental validation of theoretical claims it makes. It would be nice to at least have minimal experiments to support the results.
- This work makes few connections to practical settings, such as how their claims might account for empirical shortcomings of LLMs, and it does not discuss the implications of their results for well-known algorithmic tasks. Stronger statements of the form “UHAT Autoregressors belong to class X and therefore cannot perform task Y from a broader class” could improve the paper.

**Questions:**

1. [1] and [2] are highly relevant and are not cited in the related works. They investigate related topics namely how Transformers can express weighted/probabilistic automata/grammars. The latter paper also works on a notion quite similar to what you define as "State Encoders" through a notion they define as "Simulation".
2. Do the authors see any relationship between their work and work on "Generation in the limit"[3]? This could equally be an interesting direction for discussion.
3. Could the authors put a section "Contributions" with bullet points or something similar in the introduction? It was hard to parse what was done in the paper vs previous work when reading.
4. It would be beneficial to clarity to add brief proof sketches in the main text for (at least) the most technical theorems instead of simply deferring to the appendix.
5. I quite like Figure 1 and think it does a good job summarizing the results. However, the upwards arrows are hard to parse in terms of direction of inclusion, it was not immediately obvious for me what it meant. I feel putting a $\subseteq$ or similar might be clearer.
6. I think it could also be helpful to have a table summarizing the main results based on assumptions made, e.g. with columns "Thm/Prop number" "Semiring" "Model Type" "Main Finding"
7. Could the authors add a section discussing implications of their results to practice and to specific task families (as mentioned in the "Weaknesses" section)?


[1] Zhao, H., Panigrahi, A., Ge, R., & Arora, S. (2023). Do transformers parse while predicting the masked word?. arXiv preprint arXiv:2303.08117.

[2] Rizvi-Martel, M., Lizaire, M., Lacroce, C., & Rabusseau, G. (2024, April). Simulating weighted automata over sequences and trees with transformers. In International Conference on Artificial Intelligence and Statistics (pp. 2368-2376). PMLR.

[3] Kleinberg, J., & Mullainathan, S. (2024). Language generation in the limit. Advances in Neural Information Processing Systems, 37, 66058-66079.

---

> ### Author Response · Authors · 2025-11-21
> **Rebuttal by Authors, Part 1**
>
> # Response to Reviewer A2Nv
>
> Thank you very much for your review.
>
> > This work makes few connections to practical settings, such as how their claims might account for empirical shortcomings of LLMs, and it does not discuss the implications of their results for well-known algorithmic tasks. Stronger statements of the form “UHAT Autoregressors belong to class X and therefore cannot perform task Y from a broader class” could improve the paper.
>
> > (7) Could the authors add a section discussing implications of their results to practice and to specific task families (as mentioned in the "Weaknesses" section)?
>
> A wealth of research on transformers as classifiers (Strobl et al. 2024) has clarified the abilities and limitations of transformers on many algorithmic tasks, but it was previously unclear whether this extended to the autoregressive setting. Our results suggest that we can usually expect results on the expressivity classifiers to apply to autoregressors. We have tried to make this clearer in the revision. Some points that can be made are:
>
> - With softmax attention, autoregressive transformers should fail at recognizing the word problem for $S_5$, Boolean formula evaluation, and other algorithmic tasks difficult for softmax transformer classifiers.
> - With hard attention, autoregressive transformers should fail at recognizing $\mathsf{PARITY}$, $\mathsf{MAJORITY}$, $\mathsf{DYCK}$-$\mathsf{1}$, and other algorithmic tasks difficult for hard-attention transformer classifiers.
>
> Additionally, Section 6.3.3 makes a subtle point about practical settings; we clarified the separation between classification and autoregression in the experiments of Yang et al (2025), who found that with $k$ layers, a transformer classifier could recognize $k$ blocks of alternating symbols, but an autoregressor could recognize $k+2$.
>
> > The paper has no experimental validation of theoretical claims it makes. It would be nice to at least have minimal experiments to support the results.
>
> We agree that an empirical validation would be a valuable addition to the paper.
>
> In fact, we have already run a set of preliminary experiments to investigate this. Our initial findings were mixed:
> 1. On one hand, we found that non-star-free languages are indeed learned more poorly, which supports the theoretical impossibility from previous work.
> 2. On the other hand, the empirical difference between languages definable and not definable in $\mathsf{TL}[\mathbf{P}]$ was not clear-cut, and these results were not conclusive.
>
> We concluded that these initial experiments are not yet comprehensive enough to include. The ambiguity in the $\mathsf{TL}[\mathbf{P}]$ results suggests that designing a proper experiment to isolate these properties is a significant challenge that requires deeper investigation.
>
> Therefore, we have decided to keep the paper focused on its theoretical contributions, as these preliminary results are not yet ready for publication.
>
> > (3) Could the authors put a section "Contributions" with bullet points or something similar in the introduction? It was hard to parse what was done in the paper vs previous work when reading.
>
> We have revised the introduction to walk through the contributions of the paper hopefully more clearly.

---

> > ### Author Response · Authors · 2025-11-21
> > **Rebuttal by Authors, Part 2**
> >
> > > (4) It would be beneficial to clarity to add brief proof sketches in the main text for (at least) the most technical theorems instead of simply deferring to the appendix
> >
> > We agree and have added proof sketches for several proofs, as noted in the list of changes.
> >
> > > [1] and [2] are highly relevant and are not cited in the related works. They investigate related topics namely how Transformers can express weighted/probabilistic automata/grammars. The latter paper also works on a notion quite similar to what you define as "State Encoders" through a notion they define as "Simulation".
> >
> > > Do the authors see any relationship between their work and work on "Generation in the limit"[3]? This could equally be an interesting direction for discussion.
> >
> > Thank you for the discussion and references. We have added a discussion of simulation of automata [2] into the Related Work section. However, we think that simulating context-free grammars [1] and generation in the limit [3] are intriguing but somewhat tangential to the questions we are discussing here.
> >
> > > I quite like Figure 1 and think it does a good job summarizing the results. However, the upwards arrows are hard to parse in terms of direction of inclusion, it was not immediately obvious for me what it meant. I feel putting a $\subseteq$ or similar might be clearer.
> >
> > We actually did originally write $\subsetneq$, but they don't stretch as well as arrows, so it can be harder to see what they are relating. Arrows are consistent with the use of arrows for functions: an arrow from class A to class B means that there is a mapping from A to B that preserves the defined language.
> >
> > >I think it could also be helpful to have a table summarizing the main results based on assumptions made, e.g. with columns "Thm/Prop number" "Semiring" "Model Type" "Main Finding"
> >
> > This information is in Figure 1 already, but we are open to further suggestions about how to make Figure 1 more clear.
> >
> > > I have given a 6 and will increase my score to an 8 if the authors address points 3, 4 and 7 in the Questions/Comments section.
> >
> > We believe we have addressed these three points, and thank you for your reconsideration!
> >
> > Strobl et al. 2024: What Formal Languages Can Transformers Express? A Survey. TACL. arXiv: 2311.00208

---

> > > ### Comment · Reviewer_A2Nv · 2025-11-24
> > > **Official Comment by Reviewer A2Nv**
> > >
> > > Thank you for your detailed response. Given the addition of a detailed Discussion section and the addition of proof sketches to the main document, I will raise my score to an 8. Although this paper is solely a theoretical contribution, I believe that the novel theoretical results are strong enough to stand on their own.

---

> > > > ### Author Response · Authors · 2025-11-24
> > > >
> > > > Thank you very much!

---

### Official Review · Reviewer_uL8H · 2025-11-01

**Soundness:** 3
**Presentation:** 2
**Contribution:** 2
**Rating:** 6
**Confidence:** 3

**Summary:**

This paper presents a number of theoretical results concerning the expressivity of Transformer models, in particular in relation with counter-free, weighted Deterministic Finite Automata (DFA) and Nondeterministic Finite Automata (NFA). The central innovation consists in considering a setup that is closer to real-world usage of transformer models. In fact, while the results are still largely limited to Unique Hard Attention Transformers (UHAT), but the authors consider their use a autoregressive token generator (language models), rather than just a Boolean classifier (for language recognition). The proof techniques rely on establishing a mapping between UHAT and Linear Temporal Logic, and then proving results for LTL. The paper shows that, while some equivalences established for the Boolean classifier setting extend to the autoregressive one, other equivalences instead break.

**Strengths:**

I should start by cautioning that my expertise with the topics discussed here is limited, though I agree that understanding the expressivity of transformer models is an important research direction, given their role in powering LLMs and other modern AI advancements, even close to a decade after their initial introduction.

In my opinion, the strongest merit of the work is that of showing how results obtained in the Boolean classifier setup do not necessarily port to the autoregressive setup, which is indeed closer to how transformers are used, at least in LLMs. A natural next step would consist in applying the same treatment to SoftMax Attention Transformers.

**Weaknesses:**

While the main result is in my opinion that of showing a discrepancy between the Boolean classifier and autoregressive setup, most of the paper is devoted to proving that many equivalence results hold in both setting, thus somewhat reducing the novelty of most contributions in the work. The broke equivalence also seems to apply rather peculiar configurations (subsets of LTL).

Additionally, while considering the autoregressive setup is a step toward making the analysis more practically relevant, the work still introduces several simplifications over transformers as they are actually employed in the real world.

A few minor points:

* Some acronyms are introduced considerably later than when they are used, which decreases readability
* At lines 119-124, it would be worth point out the connection between normalized weighted languages and discrete probability distributions
* While definition 4.2 considers multiple possible suffixes, several sections in the paper appear to focus on estimating just the next token distribution, which creates confusion while reading

**Questions:**

* How challenging would it be, in your estimate, to adapt your analysis to SMATs?
* The subsets of LTL that cause the equivalence to break appear to be linked to specific operators: does that provide insights in terms of either expressiveness of complexity?

---

> ### Author Response · Authors · 2025-11-21
> **Rebuttal by Authors**
>
> # Response to Reviewer uL8H
>
> Thank you very much for your review.
>
> > While the main result is in my opinion that of showing a discrepancy between the Boolean classifier and autoregressive setup, most of the paper is devoted to proving that many equivalence results hold in both setting, thus somewhat reducing the novelty of most contributions in the work
>
> The research question that this paper set out to answer was whether existing results on language recognition (classifiers) carry over to the more realistic setting of language modeling (autoregressors). An affirmative answer is a novel result, because it tells us something about language models that we did not know before. Moreover, it is a good result, insofar as it validates existing work on language recognition. A negative answer is also a novel result. The answer we found was "both yes and no", and we consider the "yes" to be just as novel and interesting as the "no."
>
> We have revised the introduction to make this point more clearly.
>
> > Additionally, while considering the autoregressive setup is a step toward making the analysis more practically relevant, the work still introduces several simplifications over transformers as they are actually employed in the real world.
> >
> > How challenging would it be, in your estimate, to adapt your analysis to SMATs?
>
> This paper re-examines all exact equivalences between transformer encoders (i.e., excluding chain-of-thought or padding) and formal models. As such, we followed the simplifications made in those results and did not introduce any new simplifications. In doing so, we covered not only unique-hard attention, but also softmax attention with two different notions of fixed-precision. (These results were present in the original submission, and we decided post-submission to change the title of the paper accordingly.)
>
> As for further results on SMATs (e.g., with log precision), there aren't yet (to our knowledge) any exact characterizations of transformer classifiers. Given such a characterization, we think our theoretical framework could be adapted to analyze the differences between the classification and autoregression settings. Presumably, the notion of "state" would have to be extended from a finite set to a set of size $O(\log n)$.
>
> There are existing upper and lower bounds on log-precision SMATs, and we leave adaptation of these bounds to autoregressors for future work. In particular, log-precision transformers as classifiers are in $\mathsf{TC}^0$ (Merrill & Sabharwal, 2023), and we conjecture that log-precision transformers as autoregressors remain in $\mathsf{TC}^0$.
>
> >The subsets of LTL that cause the equivalence to break appear to be linked to specific operators: does that provide insights in terms of either expressiveness of complexity?
>
> These specific operators correspond to different attention patterns in transformers; separations of fragments of $\mathsf{LTL}$ imply separations of transformer variants. For instance, Prop. 6.8 shows that fixed-precision softmax transformers ($\mathsf{TL}[\mathbf{H}]$) gain some additional ability in expressing local dependencies in the autoregressive setting (autoregressive $\mathsf{TL}[\mathbf{H}]$).
>
> For any kind of attention variant that is able to express strictly local dependencies, we believe the equivalences will not break. For instance, ALiBi can be used to simulate the $\mathbf{Y}$ operator (Yang et al, 2025), suggesting that transformer classifiers and autoregressors using ALiBi will be equivalent in expressivity.
>
> Merrill & Sabharwal, 2023: A Logic for Expressing Log-Precision Transformers. NeurIPS. arXiv:2210.02671
>
> Yang et al, 2025: Knee-Deep in C-RASP: A Transformer Depth Hierarchy. NeurIPS. arXiv:2506.16055

---

> > ### Comment · Reviewer_uL8H · 2025-11-27
> > **Thanks for the reply**
> >
> > ...Though when I managed to read it was too late to update my review. My intention in any case is keep my (already positive) score, though my opinion of the works has improved a bit after your clarification.

---

### Author Response · Authors · 2025-11-21
**List of Changes**

Dear all,

Thank you very much for your reviews. Find below a list of all changes made to the draft. In addition, we have highlighted all changes made using red text in the PDF.

# List of changes

- Since the paper deals with soft-attention transformers as well as hard-attention transformers, we have retitled the paper to "Probability Distributions Computed by Transformer Language Models".
- Refocused the introduction to clarify the narrative of the paper and its results.
- Expanded the Related Work section and moved it to the end of the paper (Section 6).
- Section 5 (Expressivity Results) has been reorganized. Whereas before we discussed classifiers, then autoregressors, then various other formalisms, now we discuss the real semiring (5.2), then the Boolean semiring (5.3). This mirrors Figure 1 more closely.
- Corrected Cor 5.4 and Figure 1 to say that the formalisms in question are incomparable.
- Theorem 5.6 (formerly Theorem 6.2) is now stated for the Boolean semiring only, not a general semiring. This eliminates the awkward condition that the translations in this theorem preserve the support of the weighted language. We think the new theorem is cleaner and supports the message of the paper better (that it is autoregression, as much as weighting, that makes the language modeling setup different from language recognition).
- Additionally, the former Lemmas 6.1 and 6.2, which served as the proof sketch for Theorem 5.6 (formerly Theorem 6.2), have been relegated to the appendix (as Lemmas B.1 and B.2) and replaced with a more readable sketch.
- Proposition 5.7 (formerly Corollary 6.3) has been rewritten to reflect the fragments of temporal logic we use, and a short motivation has been added above it.
- The proof of Proposition 5.9 (in Appendix D) has been rewritten without using semigroup theory.
- Replaced the Conclusion with a Discussion (Section 7) addressing implications for applications and future work.
- Many small clarifications and corrections.

---

### Meta-Review · Area_Chair_LLGw · 2026-01-07

**Summary:**

## Summary
This paper studies the expressive power of Transformers as formal language recognizers. In contrast to most existing results studying Transformers as classifiers for Boolean-weighted languages, the paper considers four settings, dividing the cases along two axes: Boolean-weighted vs real-weighted languages and classifiers vs autoregressors. The paper shows a number of results, also drawing connections and establishing equivalence between Transformers, counter-free deterministic finite automata, and linear temporal logic (and its fragments). In short, the paper shows that while autoregressors are at least as expressive as classifiers in Boolean-weighted languages (where equivalence holds for rightmost unique-hard attention), autoregressors and classifiers are not comparable in real-weighted languages.

## Reviewer Concerns
Major concerns raised by reviewers can be summarized as follows.
- **Softmax and PEs**. Reviewers uL8H and hwiM asked if any results could be extended to softmax attention, with a positional encoding as well.
- **No experiments**. Reviewers DAYT and A2Nv expressed concerns that the results are purely theoretical and empirical validation is missing.
- **Incremental results**. Reviewer hwiM claimed that some results are incremental, based on existing results.
- **Connection to practice**. Reviewer A2Nv asked if the results can be connected to ability/inability of transformers in solving certain tasks.
- **Accessibility/readability**. Reviewers A2Nv, DAYT, and hwiM commented that the paper is dense and difficult to read, with limited accessibility for readers outside the TCS community.
- **Writing suggestions**. Reviewers made suggestions on writing, such as discussion on connection to practice and future work, adding a contribution section in the introduction, adding proof sketches and a summary table, etc. Some reviewers pointed out missing relevant results and minor typos.

**Reviewer Concerns:**

Based on the rebuttal, the authors’ responses can be summarized as follows.

- **Softmax and PEs**. The authors clarified that some of the stated results in the paper pertain to fixed-precision softmax attention, and also changed the paper title to reflect this. They also discussed extension to log-precision softmax attention which they leave as future work. The authors also speculated that their framework can be used to analyze different kinds of positional embeddings by leveraging their connections with different operators in temporal logic (e.g., ALiBi is connected with the $Y$ operator).

- **No experiments**. The authors briefly described inconclusive results in their experiments and explained why they were ultimately omitted.

- **Incremental results**. The authors clarified that no previous work considered what probability distributions transformer language models (autoregressors) could represent, while existing results that the reviewer pointed out focus on Boolean classifiers.

- **Connection to practice**. The authors discussed examples of tasks that cannot be recognized by autoregressive Transformers with softmax/hard attention. Some are briefly mentioned in the new Section 7.

- **Accessibility/readability**. The authors added pointers to more detailed introductions of the necessary concepts such as temporal logics and finite automata.

- **Writing suggestions**. The authors made an extensive revision and incorporated the suggestions.

**Reviewer Scores:**

The initial reviews had scores 6/6/6/4, which is already quite positive. Three positive reviewers (uL8H, A2Nv, DAYT) responded to the authors’ rebuttal and Reviewer A2Nv raised their score to 8.

**Reviewer hwiM** (score 4) was not able to respond to the authors before the discussion period closed. Given the extensive effort the authors put into the rebuttal and the revision, I believe that most of the major concerns by the reviewer were well-addressed. It is plausible to expect that the reviewer might have raised the score.

Upon reading the paper myself, I believe the paper presents strong theoretical results that expand the discussion of Transformer expressivity beyond Boolean classifiers. Although I agree with Reviewer DAYT that ICLR may not be the most natural venue for this submission, the paper is indeed an important contribution to the theory of Transformers. I am happy to recommend acceptance of this paper.

Lastly though, I must say that the paper is technical and not easily accessible to people without prior knowledge on finite automata and formal languages. I recommend the authors to revise the paper to make it more self-contained and accessible. Below are some minor/clarifying suggestions:
- Clarify the definition of $\epsilon$ (empty string) in Eq (5).
- Clarify why cfDFAs = cfNFAs for real classifiers in Fig 1.
- Clarify in Def 5.1 that the $k_i$’s are weights (constants).
- Clarify the notation $(1a)^\ast$ and $(\frac{1}{2}a)^\ast$ in Corollary 5.4 before using it.
- Line 357: “it is easy to write an LTL **autoregressor** to recognize this”?
- Line 426: by “these equivalences break” do you mean the equivalence between Transformers and $TL[P]$, or the equivalence between classifiers and autoregressors? If the authors meant the former, it seems to contradict Figure 1.

---

### Decision · Program_Chairs · 2026-01-26

Accept (Poster)